# Tall wind profile validation of ERA5, NORA3 and NEWA, using lidar observations

Etienne Cheynet[1,*], Jan Markus Diezel[1,*], Hilde Haakenstad[2], Øyvind Breivik[2,3], Alfredo Peña[4], and Joachim Reuder[1]

[1]Bergen Offshore Wind Centre (BOW) and Geophysical Institute, University of Bergen, Norway
[2]Norwegian Meteorological Institute, Norway
[3]Geophysical Institute, University of Bergen, Norway
[4]DTU Wind and Energy Systems, Technical University of Denmark, Denmark
[*]These authors contributed equally to this work.

**Correspondence:** Etienne Cheynet (etienne.cheynet@uib.no)

**Abstract.** The development of large wind turbines and airborne wind energy (AWE) systems requires reliable wind speed datasets at heights above the atmospheric surface layer. Traditional measurement approaches relying on met-masts, fall short of addressing these needs. In this study, we validate three different model-based datasets, namely the 3-km Norwegian Hindcast archive (NORA3), the New European Wind Atlas (NEWA), and ECMWF Reanalysis v5 (ERA5) using Doppler wind lidar data from several locations in Norway and the North Sea. The validation focuses on altitudes from 100 to 500 m above ground, covering the operational range of large wind turbines and AWE systems. Our findings indicate that ERA5 and NORA3 perform similarly well in offshore locations in terms of bias, correlation coefficient, root-mean-square error and Earth's mover distance. The choice of an appropriate wind speed database depends on the topography, altitude and error metrics of interest. However, NORA3 outperforms the other two models in two coastal and one complex terrain sites. In most cases, the agreement between the models and lidar measurements increases with height.

## 1 Introduction

The hub height and rotor diameter of wind turbines have been continuously increasing for the past 20 years (Jahani et al., 2022; Jiang, 2021), reaching up to 150 m and over 200 m, respectively. This trend is driven by the need to capture stronger and steadier winds to reduce the levelized cost of energy (Wiser et al., 2021). In the near future, the top tip height of such wind turbines may exceed 300 m (Rogers, 2024).

Airborne Wind Energy (AWE) systems harness wind energy using tethered aircraft operating at altitudes between 200 and 600 m. At these heights, wind speeds are generally stronger and steadier than near the surface. Since the 2010s, AWE systems have made significant advances (Vermillion et al., 2021; Fagiano et al., 2022; Eijkelhof and Schmehl, 2022). Prototypes with capacities exceeding 600 kW have been developed, and scaling to multi-megawatt systems has been proposed (Vermillion et al., 2021; Kruijff and Ruiterkamp, 2018). Despite this progress, AWE systems are still in the early stages of development compared to conventional wind turbines.

Two main concepts dominate current AWE designs: ground-generation systems and onboard-generation systems. Ground-generation systems, or "pumping mode" systems, generate energy on the ground using a winch and generator. In this case, the tethered aircraft alternates between the energy-generation and recovery phases. Aircraft for this concept include soft kites, semi-rigid, and rigid wings. Each type offers trade-offs between adaptability and durability. Onboard generation systems, in contrast, produce energy in the air using onboard turbines and transmit energy to the ground via conductive tethers. These systems typically use rigid-wing aircraft, quadrotors, or toroidal aerostats (Cherubini et al., 2015). While ground-generation systems are relatively efficient, they require advanced automation for continuous operation (Elfert et al., 2024). Onboard-generation systems are better at harnessing high-altitude winds but face challenges in weight optimisation and tether design. Soft wings are adaptable to varying wind conditions but are less durable. Conversely, rigid wings provide higher power output but have greater mechanical complexity and costs (Fagiano et al., 2022). Key challenges remain for AWE systems, including managing wind variability, tether dynamics, and autonomous operation. A major limitation lies in the reliance on oversimplified wind speed approximations, due to the lack of detailed wind speed data at altitudes above 200 m (Sommerfeld et al., 2019). Addressing this gap through tall wind profiling is essential for optimising AWE system design and unlocking their full potential for large-scale deployment.

Traditional methods for estimating horizontal mean wind speed profiles rely on mast-based measurements, which are primarily applicable to studying the atmospheric surface layer (ASL). The ASL constitutes approximately the lowest 10% of the atmospheric boundary layer (ABL). Under neutral and unstable thermal stratifications of the atmosphere, the depth of the ASL ranges from 50 m to 300 m (Hennemuth and Lammert, 2006; Pal and Lee, 2019; Davis et al., 2020, 2022). Under stable atmospheric conditions, the ASL becomes shallow, sometimes reaching a depth below 30 metres (Berström and Smedman, 1995). Traditional logarithmic and power-law mean wind speed profiles, such as those used in IEC 61400-1 (IEC, 2005), are limited to the ASL and are likely inadequate for wind turbine design with hub heights of 150 m or more (Tieo et al., 2020; Cheynet et al., 2024). For wind resource assessment, the height-dependent Weibull parameters (Kelly et al., 2014) require the characterisation of wind speed profiles above 200 m. Therefore, the development of both AWE systems and future offshore wind turbines necessitates information on the mean wind speed at heights several hundred meters above the surface.

Tall wind speed profiles, as defined here, cover at least the initial 500 m above the surface, and their characterisation remains a significant challenge (Veers et al., 2019). The term 'tall wind profile' is in line with its use in boundary-layer meteorology (e.g. Peña et al., 2014; Kelly et al., 2014). Traditional tall masts often fall short of this definition, as they are typically lower than 100 m. Only a limited number of masts exceeding 200 m exist globally (Ramon et al., 2020), and these are exclusively onshore masts (Table 1). Although tall-wind speed profiles can also be collected using manned aircraft (e.g. Zemba and Friehe, 1987) or drones (Egger et al., 2002; Reuder et al., 2009; Palomaki et al., 2017; Shimura et al., 2018), this approach has not been adopted for wind resource assessment, which requires several years of data. In offshore locations, the collection of tall wind speed profiles is further complicated by the harsh marine environment and the high costs of installation and maintenance of sensors and their supporting structure (Shaw et al., 2022).

Tall wind speed profiles are generally measured using remote sensing technologies (Emeis, 2011), such as Doppler wind radar (Lehmann and Brown, 2021), sodar (Bianco, 2011), and lidar (Pichugina et al., 2012). As commercial Doppler wind lidars

(DWLs) have become the standard instrumentation for wind energy applications, we have based our study on corresponding available lidar data sets. Commercial DWL profilers measure wind speed and direction up to approximately 300 m above the surface (Peña et al., 2009). They use the aerosol backscatter from emitted laser beams to determine the wind speed at various

heights. Scan modes such as the Doppler beam swinging (DBS) or velocity azimuth display use multiple beams at different azimuths and a fixed elevation angle to calculate the average wind speed within the encompassed scan volume, showing good performance against tall met masts (Knoop et al., 2021). Commercial DWL profilers, both ground-based and mounted on fixed platforms, have been used in wind energy research for over a decade, both onshore (Smith et al., 2006; Kumer et al., 2016; Brune et al., 2021) and offshore (Peña et al., 2009; Peña et al., 2015; Brune et al., 2021). In the 2010s, floating wind lidar profilers

deployed on buoys (Gottschall et al., 2017; Peña et al., 2022) and ships (Rubio et al., 2022) began to complement traditional offshore met-masts, offering cost reductions (Krishnamurthy et al., 2013).

Most commercial DWL wind profilers are still limited to studying the mean wind flow up to 300 m above the surface. Scanning DWLs possess a more powerful laser than lidar profilers, allowing them to collect data up to 3 km along the scanning beam under good aerosol conditions (Kumer et al., 2014). This technology, used in atmospheric research for over two decades (Pichugina

et al., 2012; Banta et al., 2013; Dias Neto et al., 2023), saw commercial adoption mainly in the early 2010s (Vasiljevic, 2014; Kumer et al., 2014). Scanning DWLs with hemispherical scanning capabilities can adjust both azimuth and elevation angles and can be set to mimic profiler modes for direct atmospheric profiling. Despite their potential, scanning lidars are underused in developing AWE systems or next-generation multi-megawatt offshore wind turbines. Notable examples of wind speed data collection with such instruments for tall wind profile analysis include Kumer et al. (2014) in coastal terrain, Reuder et al. (2024)

at the FINO1 offshore platform, Päschke et al. (2015) and Sommerfeld et al. (2019) in Germany or Mariani et al. (2020) in the Arctic.

Recently, open-source datasets from wind hindcast or reanalysis databases such as the 3-km Norwegian Hindcast archive (NORA3) (Haakenstad et al., 2021), the New European Wind Atlas (NEWA) (Hahmann et al., 2020), and ECMWF Reanalysis v5 (ERA5) (Hersbach et al., 2020) have provided model-based wind speed profiles within the first 500 m above the surface. A

hindcast is a numerical model simulation of a historical period without data assimilation, unlike a reanalysis. Reanalysis uses modern data assimilation techniques over decades, combining observations with models to create a consistent picture of weather and climate, even in areas without direct observations. NORA3 and ERA5 appear adequate for wind resource assessment and structural design, for which a climatological description of wind conditions spanning at least 30 years is necessary. Although these databases can complement measurements, they require proper validation for wider use in wind energy. Portions of these

databases have undergone validation against near-surface measurements (e.g. Ramon et al., 2019), mast measurements at levels below 100 m (Olauson, 2018; Jourdier, 2020; Solbrekke et al., 2021; Cheynet et al., 2022), or tall-masts measurements up to 200 m (Knoop et al., 2020; Gualtieri, 2021). Additional validation has been conducted using Doppler wind lidar technology for heights up to 300 m above the surface (Pronk et al., 2022; Hallgren et al., 2024). The comparison of lidar wind speed profiles with hindcast or reanalysis products at higher altitudes remains, however, limited. This motivates our study to bridge

this identified knowledge gap.

**Table 1.** Examples of tall met masts with top height above 200 m documented in Ramon et al. (2020) and in the scientific literature. None of these masts are offshore. The list includes both current and former masts.

| Tower name | Top sensor (m) | Country |
| --- | --- | --- |
| Walnut Grove | 488 | USA |
| Park Falls | 396 | USA |
| West Branch | 379 | USA |
| South Carolina | 329 | USA |
| Beijing Meteorological Tower | 325 | China |
| Gartow | 341 | Germany |
| Obninsk | 301 | Russia |
| Boulder Observatory | 300 | USA |
| Boseong | 300 | South Korea |
| Hamburg | 280 | Germany |
| Steinkimmen | 252 | Germany |
| Østerild | 250 | Denmark |
| KIT | 200 | Germany |
| Cabauw | 200 | Netherlands |

In this study, we validate datasets from ERA5, NEWA and NORA3 against measurements from DWL systems across diverse terrain locations in the North Sea and Norway to evaluate their accuracy in capturing tall wind speed profiles. We also aim to quantify the performance of these databases as a function of altitude using multiple error metrics. The selection of different sites provides an opportunity to examine the impact of topography on local wind conditions. The novelty of this study lies in the

gathering of lidar tall wind speed profiles and their comparison with various hindcast and reanalysis wind databases at altitudes up to 500 m, a topic that has received little attention to date. Moreover, this study explores the use of wind-energy-related metrics, such as the capacity factor of hypothetical modern wind turbines and AWE systems, to evaluate the wind speed databases. This research is believed to be valuable to both wind energy and wind engineering, as tall wind speed profiles can be used for both wind resource assessment (Schelbergen et al., 2020) and the analysis of wind loading on structures (Kent et al., 2018).

This study is organised as follows: section 2 introduces the datasets extracted from the ERA5, NORA3 and NEWA databases, as well as the DWL data. Section 3 presents the metrics used for the error analysis and the data processing to collocate the modelled and measured data in space and time. Section 4 examines how site selection and measurement height affect the errors between wind atlases and remote-sensing measurements. Our findings underscore NORA3's and ERA5's reliability offshore, especially above 100 m. However, in coastal areas and complex terrains, regional model-based datasets and microscale wind

models may be required to accurately capture local wind conditions. Section 4 also presents how the selection of the wind atlas impacts the estimation of the capacity factor of modern wind turbines and AWE systems. Finally, section 5 discusses the need

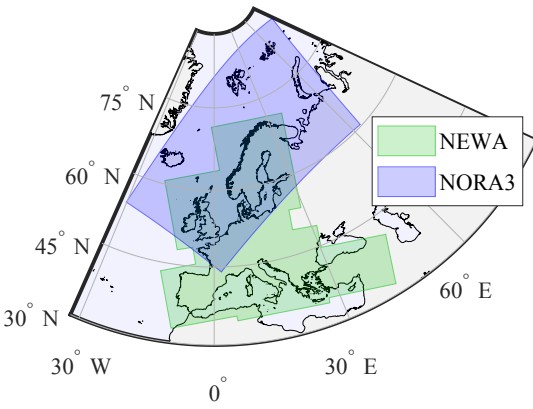

**Figure 1.** Illustration of the regions covered by NEWA and NORA3. ERA5 has global coverage and is thus not shown here.

for more powerful DWL profilers and the complementary role of mesoscale and microscale simulations for improved wind resource assessment.

## 2 Data collection and wind models

This section outlines the wind models and measurement campaigns considered to assess tall wind speed profiles. NEWA, NORA3, and ERA5 wind atlases are introduced with their spatial and temporal resolutions. Five lidar campaigns across offshore, coastal, and complex terrain sites provide validation data for the analysis.

### 2.1 Model data

A state-of-the-art wind atlas is defined herein as a climate dataset that provides the mean wind speed and mean wind direction at multiple heights above the surface. It provides a horizontal spatial resolution on the kilometre scale, a time resolution of 1 h or finer, and temporal coverage of at least 30 years. The definition of a wind atlas employed here relates to a hindcast or reanalysis database that is usable for wind resource assessment or the design of wind energy systems, including extreme value analysis. In this study, they are regional downscaling products of the ERA5 reanalysis (Hersbach et al., 2020), covering overlapping areas in Europe (Fig. 1). Consequently, ERA5 data are also included in this analysis. Although wind atlases are sometimes defined as databases with microscale spatial resolution finer than 1 km, we choose to adopt a broader definition that includes model-based wind data with a kilometre-scale resolution. Hereinafter, $z$ denotes the height in metres above the surface, and $\overline{u}$ represents the horizontally averaged mean wind speed at height $z$.

The ERA5 reanalysis product from the European Centre for Medium-Range Weather Forecasts (ECMWF) superseded ERA-Interim in 2019 (Dee et al., 2011). ERA5 offers climate data with global coverage, a horizontal spatial resolution of approximately 31 km and a hourly output, extending from 1940 onward. As a reanalysis product, ERA5 uses a 4D-Var data assimilation scheme (Courtier et al., 1994) within a 12-hour assimilation window, incorporating both in-situ measurements

**Table 2.** Metadata of the wind atlas data sets. Only z-levels above 100 m are shown and used hereinafter. ERA5 data were collected using the 100 m z-level with additional height levels retrieved using pressure levels and the geopotential height. $\Delta h$ is the horizontal spatial resolution and $\Delta t$ is the temporal resolution in minutes.

|  | $\Delta h$ (km) | $\Delta t$ (min) | z-levels (m) |
|---|---|---|---|
| NORA3 | 3 | 60 | 100, 250, 500, 750 |
| NEWA | 3 | 30 | 100, 150, 200, 250, 500 |
| ERA5 | 31 | 60 | 100, geopotential heights |

and satellite observations (Hersbach et al., 2020). The ERA5 temporal resolution and coverage are valuable for wind energy research globally (Olauson, 2018), including the analysis of annual and decadal wind variability and potential trends due to climate change (Chen, 2024; Antonini et al., 2024; Martinez and Iglesias, 2024). Researchers and engineers in the wind energy sector increasingly rely on ERA5 for both historical analysis and future project planning (Olauson, 2018; Gualtieri, 2021; Hayes et al., 2021).

NEWA is an open-access European wind atlas released in 2019 and developed through collaboration among 30 European academic and industrial partners (Hahmann et al., 2020; Dörenkämper et al., 2020). NEWA uses the Weather Research and Forecasting model and utilises the ERA5 reanalysis as forcing, without further data assimilation. NEWA covers the period from 1989 to 2018 and aims to provide a detailed wind climatology of Europe with a spatial resolution of approximately 3 km and a temporal resolution of 30 min. In this study, data on mean wind speed and direction from the mesoscale output of NEWA were retrieved at eight altitudes, ranging from 10 m to 500 m above mean sea level.

NORA3 is a regional downscaling of the ERA5 reanalysis and utilises the HARMONIE-AROME non-hydrostatic regional numerical weather prediction model for its production (Haakenstad et al., 2021). The downscaling consists of nine-hour short integration runs initiated every six hours, employing the last run as the initial state for the new cycle. Surface observations in the CANARI-OI-Main assimilation system adjust the first-guess (Giard and Bazile, 2000; Taillefer, 2002), with ERA5 forcing applied in the free atmosphere following the boundary relaxation method (Radnoti, 1995; Termonia et al., 2018). NORA3 is built on NORA10's legacy, which has been used by the offshore industry in the North Sea for over a decade (Furevik and Haakenstad, 2012). Validations of the NORA3 database against atmospheric data include comparisons with wind measurements from oil and gas platforms, as well as offshore masts in the North Sea and Norwegian Sea (Solbrekke et al., 2021). NORA3 provides wind data once per hour with a 3 km spatial resolution. A specific subset, released in 2021 by the Norwegian Meteorological Institute, has been selected for this study. This subset presents mean wind speed and direction data at seven heights, from 10 to 750 m above sea level, extending from 1961 onward, as reflected in the database status at the end of 2024. Table 2 details the metadata for the different models, illustrating the variability in spatial and temporal resolutions, as well as available height levels above 100 m among NORA3, NEWA, and ERA5.

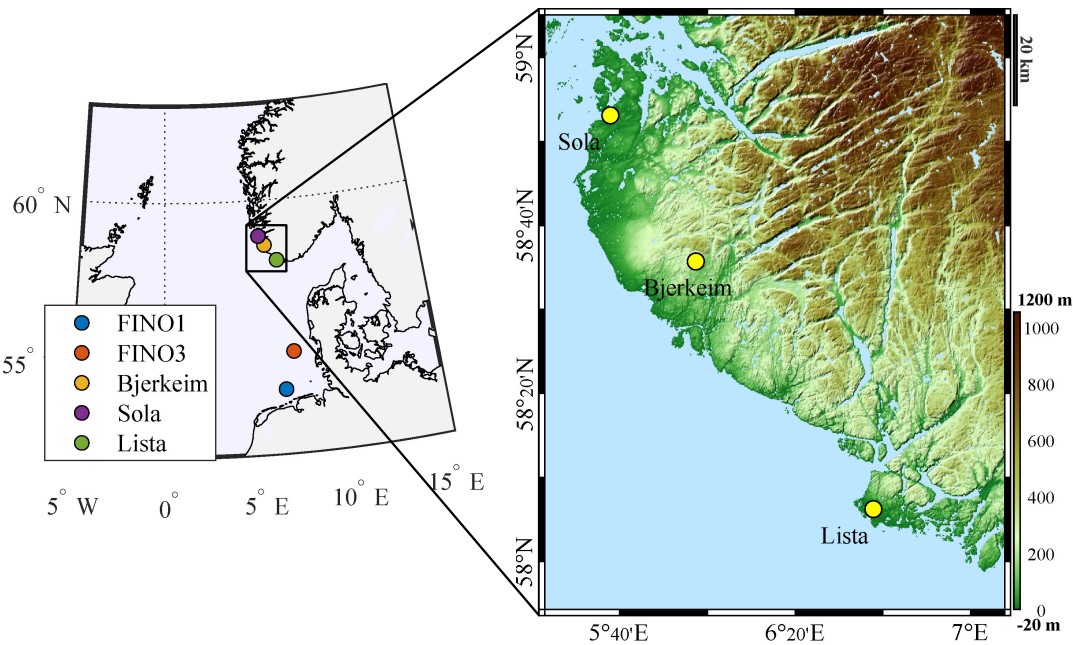

**Figure 2.** Locations of the five measurement sites in the North Sea and along the Norwegian coast (left panel), with a detailed view of the onshore sites (right panel). The right panel includes a digital elevation model of the three onshore measurement sites, generated using the toolbox by Beauducel (2024).

## 2.2 The measurement sites

The measurement data were collected by reference DWL instruments within the area covered by ERA5, NORA3, and NEWA (Fig. 2). Two lidar campaigns were conducted in the marine ABL (FINO1 and FINO3 platforms), two others at coastal sites (Sola and Lista airports in Norway), and one in complex mountainous terrain (Bjerkreim, Norway), characterized by rocky terrain and
sparse vegetation. FINO1's proximity to wind farms enables an examination of discrepancies between lidar measurements and wind atlases due to wake effects. Analysing data from both the FINO1 and the FINO3 platforms highlights the challenges of assessing wind resources in such areas and underscores the need for cautious application of wind atlases near wind farms.

    Table 3 summarises the locations and measurement periods of the five campaigns selected for the validation of wind atlases. Figure 2 geolocalises the five measurement campaigns and provides a close-up of the three onshore locations. The offshore sites
are situated more than 40 km away from the coast. The coastal sites are located inland only a few kilometres from the shore and are characterised by sharp roughness changes as the terrain transitions from open water to flat, agricultural land with sparse vegetation. These abrupt roughness changes generate internal boundary layers, which can be challenging to capture accurately with hindcast and reanalysis databases. The complex site Bjerkreim is mountainous, with steep slopes and limited vegetation or trees. While distinct from the fjord-like landscapes found in other parts of Norway, this complex terrain features significant
elevation changes that contribute to non-homogeneous wind conditions, particularly within the atmospheric surface layer.

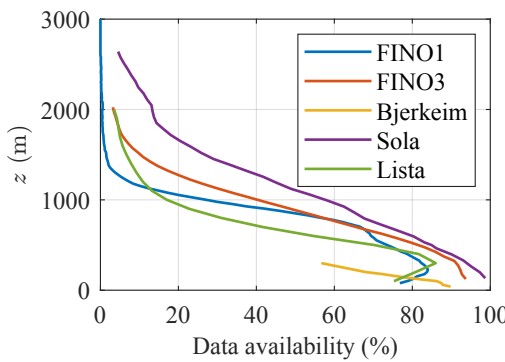

**Figure 3.** Data availability during the five measurement campaigns as a function of the altitude, based on the Carrier-to-Noise Ratio of -27.5 dB for the scanning lidars and -22 dB for the lidar profilers.

**Table 3.** Metadata of datasets from five lidar measurement campaigns. The lidar range gate denotes the along-beam spatial resolution.

|  | FINO1 | FINO3 | Lista | Sola | Bjerkreim |
|---|---|---|---|---|---|
| Latitude (N) | 54.015 | 55.195 | 58.104 | 58.885 | 58.595 |
| Longitude (E) | 6.588 | 7.158 | 6.631 | 5.631 | 5.955 |
| Terrain Type | Offshore | Offshore | Coastal | Coastal | Complex |
| Device | Windcube 100S | WLS70 | WLS70 | Windcube 100S | WindCube V1 |
| Start Date | 01.06.2015 | 10.09.2013 | 20.11.2020 | 04.03.2013 | 30.03.2010 |
| End Date | 05.10.2016 | 06.10.2014 | 06.09.2021 | 30.06.2013 | 06.05.2010 |
| Hours Collected | 1353 | 7997 | 6912 | 525 | 799 |
| Min height (m) | 78 | 125 | 100 | 133 | 40 |
| Max height (m) | 3528 | 2025 | 2000 | 2641 | 300 |
| Range Gate (m) | 25 | 50 | 100 | 66 | 20 |

The first measurement campaign took place in Bjerkreim, Norway, from March to May 2010. A WindCube V1 DWL (Leosphere) was deployed to capture wind profiles from 40 m to 300 m above the surface. Approximately 799 hours of mean wind speed data were collected with a temporal resolution of 10 minutes. The records were subsequently validated against sodar measurements taken 300 m from the lidar's location.

The second measurement campaign was conducted at Stavanger Airport Sola, Norway, from March to June 2013 (Kumer et al., 2014). The objective was to evaluate the capability of a long-range scanning pulsed lidar instrument in measuring mean wind speed at altitudes beyond the reach of traditional meteorological masts. A first-generation WindCube 100S lidar was deployed, operating in DBS scanning mode and capturing 10-minute mean wind speed profiles at a minimum height of 133 m above the surface. This campaign resulted in the collection of approximately 525 hours of data. The lidar measurements were

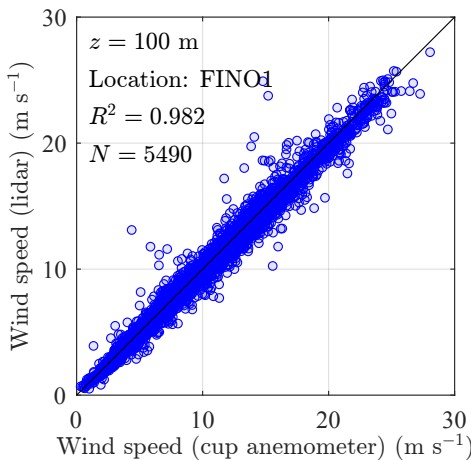

**Figure 4.** Comparison of 10-minute mean wind speed measurements recorded by the cup anemometer and the WindCube 100S at 100 m above sea level, collected between June 2015 and October 2016 during the OBLEX-F1 campaign at the FINO1 platform (5490 samples).

compared against radiosonde data, demonstrating a strong correlation with a Pearson coefficient $R > 0.95$, which improved with increasing altitude (Kumer et al., 2014).

The third measurement campaign took place at the offshore platform FINO3, located at $55.195°$N, $7.158°$E in the North Sea, about 80 km off the Danish coast. The platform hosted a Leosphere WLS70 DWL profiler (Cariou et al., 2009) which operated from September 2013 to October 2014. This resulted in the collection of approximately 7997 hours of data with a temporal resolution of 10 minutes. The lowest measurement height of 125 m above sea level ensured no significant mast-induced flow distortion. The collected wind speed and direction data were validated and detailed in Peña et al. (2015).

The fourth measurement campaign was conducted at the FINO1 offshore platform located at $54.015°$N, $6.588°$E in the southern North Sea, approximately 45 km north of the island of Borkum, Germany. The platform hosted a Windcube 100S (Leosphere) long-range scanning DWL from June 2015 to October 2016 (Reuder et al., 2024). The lidar operated in DBS scan mode with a fixed elevation angle of $70°$, performing scans twice per hour to collect 10-minute mean wind speed profiles. In total, approximately 1353 hours of data were recorded, with the lowest measurement height at 78 m above sea level. A preliminary comparison with a reference cup anemometer mounted at the top of the FINO1 mast at 100 m above sea level (Fig. 4) showed excellent agreement, with a squared Pearson coefficient ($R^2$) of 0.98, similar to results from a study conducted at the FINO3 platform (Peña et al., 2015).

The fifth measurement campaign took place at Farsund Airport (Lista, Norway) from November 2020 to September 2021, focusing on the study of the ABL for airborne wind energy applications. A WindCube WLS70 lidar was deployed and operated in DBS mode with measurement heights ranging from 100 to 2000 m above the surface. The measurements resulted in the collection of approximately 6912 hours of data with a temporal resolution of 10 minutes.

It should be noted that the measurements at Bjerkreim and Sola were conducted over short time periods and are therefore not
representative of the typical, annual wind variations at these sites. Instead, the data should be interpreted within the context of
this study, which aims to compare tall wind speed profiles from wind atlases with lidar observations.

Figure 3 displays the lidar data availability as a function of the altitude, which decreases substantially above 500 m due to low
clouds and low aerosol content. The data availability is defined here by the carrier-to-noise ratio (CNR). Data with a CNR below
-27.5 dB for the scanning lidar instrument and below -22 dB for the DWL profiler were excluded. A threshold of -27.5 dB allows
for increased data availability and is empirically robust for scanning Doppler wind lidar (Cheynet et al., 2017). For simplicity,
specific methods, e.g. those presented by Beck and Kühn (2017); Valldecabres et al. (2018); Cheynet et al. (2021) or Duscha
et al. (2023) to recover data with even lower CNR are not considered here, as only the mean wind speed was of interest.

All lidar instruments operated in DBS mode with an opening angle of 30 degrees. This mode reconstructs horizontal wind
speed and direction from the radial velocities of four beams. It also incorporates two volume averaging effects: one over the
205 range gate of each beam and another across the scan horizontal area, which increases with height. The DBS mode assumes
horizontal homogeneity of the mean flow, a condition typically met in flat and homogeneous terrain but violated in highly
complex terrain (Pauscher et al., 2016) or near obstacles, such as in the wake of a wind turbine. As the altitude increases, the
flow tends to become more homogeneous and horizontal (Emeis, 2013), making DBS scanning suitable for measuring tall wind
speed profiles.

## 3 Methods

This section outlines the methods used to compare lidar wind speed data with predictions from NORA3, NEWA, and ERA5. It
covers the four error metrics employed, data preprocessing techniques, and the evaluation of power curves and capacity factors
for wind turbines and airborne wind energy systems.

### 3.1 Error metrics

To quantify discrepancies between mean wind speed data obtained from lidar instruments and those predicted by wind atlases,
four metrics are employed: the $R^2$ coefficient, the bias, the Root-Mean-Square Error (RMSE), and the first Wasserstein distance,
also known as the Earth Mover's Distance (EMD). The $R^2$ coefficient measures the linear correlation between the measured
and modelled wind speeds. The bias quantifies the average difference between the predicted and observed values, providing a
measure of systematic error. The Root-Mean-Square Error (RMSE) quantifies the average magnitude of the error. The EMD
quantifies the dissimilarity between two probability distributions, making it well-suited for analysing wind atlases that represent
the climatology of a site in terms of the probability distribution of mean wind speed and direction (Hahmann et al., 2020). For
brevity, only the EMD equation is introduced, as the equations for the $R^2$ coefficient, RMSE, and bias are assumed to be familiar
to the reader. For one-dimensional distributions, the EMD can be represented by the area between two cumulative distribution

functions, $F_1$ and $F_2$:

$$\text{EMD} = \int\limits_{-\infty}^{+\infty} |F_1(x) - F_2(x)| \, \mathrm{d}x. \tag{1}$$

To complement these metrics, the Taylor diagram (Taylor, 2001) provides a summary of model performance by integrating the correlation coefficient, standard deviation, and RMSE into a single plot. This graphical representation is particularly useful for comparing multiple models against observed data in a visually intuitive way.

## 3.2  Data preprocessing

Wind atlases data were initially interpolated from their original horizontal grid to the GPS coordinates of the lidar campaign locations at each vertical height level. The interpolation scheme follows the method described in Amidror (2002), specifically a linear scattered data interpolation, as the data from the wind atlases are not necessarily on a Cartesian grid.

The NORA3 and NEWA datasets are provided at specific height levels (Table 2). The ERA5 data were collected at five pressure levels (1000, 975, 950, 925, and 900 hPa), as well as at 10 and 100 m above the surface. Combining height and pressure levels ensured robust data retrieval and high vertical resolution. The pressure levels were converted into geopotential height levels using the geopotential variable available in the ERA5 database. Geopotential height differs slightly from geometric height as it accounts for variations in Earth's gravity, while geometric height is the actual vertical distance. This study focuses on wind speed data within the first 500 m above the surface, where the two heights are approximately equivalent.

The wind atlas data were linearly interpolated to the measurement heights of each lidar range gate. Since each lidar has different range gates, the comparison cannot always be conducted at the exact same altitude across all sites. Alternative non-linear interpolation schemes were also tested, such as spline, piecewise cubic Hermite interpolating polynomial methods, and the modified Akima method (Akima, 1974), but they yielded similar results while being less robust than the linear interpolation. Appendix A presents the error metrics quantified at specific heights and as vertical profiles using a non-linear regression. The non-linear regression shows minor differences in the error metrics compared to linear interpolation but does not change the conclusions of the study. The comparison in appendix A supports our decision to use linear interpolation for additional height levels in this study.

Following the spatial collocation, the model data were then temporally collocated with the lidar data. The lidar data consist of 10-minute averaged time series, while the model data are provided at 60-minute or 30-minute resolutions. The first approach (Approach A) involves a temporal interpolation of the model data to align with the 10-minute averages from the lidar. An alternative approach (Approach B) would be to interpolate the lidar data to an hourly timestep. However, Approach B is more complex than Approach A and did not yield significant differences. Therefore, for simplicity and to avoid overprocessing the data, Approach A was adopted. Finally, for the lidar data, the initial outlier detection and removal method relied on CNR threshold values of -27.5 dB for the scanning DWL and -22 dB for the DWL profiler. This method was found to be sufficiently effective, eliminating the need for additional outlier tests.

## 3.3 Power curves and capacity factor

This subsection outlines the methodology used to evaluate how data from different wind atlases influence the estimated capacity factors of wind turbines and AWE systems at the five selected sites. The wind turbine models examined include the NREL 5 MW (90 m hub height, 126 m rotor diameter), IEA 15 MW (150 m hub height, 240 m rotor diameter), and NREL 18 MW (156 m hub height, 263 m rotor diameter), the details of which are provided by NREL (2020). The power curves for these turbines are defined by a rated wind speed typically between 10 and 12 m s$^{-1}$, a cut-in speed of 3 to 4 m s$^{-1}$ and a cut-out speed of approximately 25 m s$^{-1}$. Capacity factor calculations in this study are based on wind speed at hub height. While modelling rotor-averaged (or equivalent) wind speed, which accounts for shear, turbulence intensity, and wind veering (Wagner et al., 2009; Antoniou et al., 2009; Murphy et al., 2019) could yield more realistic capacity factor estimates, such an analysis falls beyond the scope of this work.

The power curve for AWE systems depends on flight height and trajectory, complicating the estimation of annual energy production and capacity factors. To address this challenge, various approaches are possible, such as clustering methods for faster computation of AWES production (Schelbergen et al., 2020; Sommerfeld et al., 2023) or simplified power curves (Eijkelhof and Schmehl, 2022; Ranneberg et al., 2018). In this study, we opted to use simplified power curves as we focus primarily on validating tall wind speed profiles using scanning lidar instruments, without delving into AWE systems flight trajectory optimisation. Two simplified power curves for AWE systems are considered: one developed for a 3 MW rigid body (Eijkelhof and Schmehl, 2022) and one for a smaller 100 kW AWE system (Ranneberg et al., 2018). The power curves of the 3 MW and 100 kW systems are displayed in the middle and bottom panels of Fig. 5, respectively. These power curves represent a balance between simpler ones, such as those used in Vos et al. (2024), where the rated power remains constant above the rated wind speed, and more advanced path-dependent approaches studied in Eijkelhof and Schmehl (2022) or Sommerfeld et al. (2023).

In the middle panel of Fig. 5, we smoothed the 3 MW power curve using non-linear regression with smoothing splines, leading to cut-in and cut-out operating wind speeds of 9.0 and 29.9 m s$^{-1}$, respectively. These values slightly differ from those in the study by Eijkelhof and Schmehl (2022), which were based on ten optimised flight paths. Notably, we adopted a lower cut-in wind speed of 9 m s$^{-1}$, compared to their 10 m s$^{-1}$. This distinction may be significant in the North Sea, where median wind speeds at 250 m above the surface typically range from 9 to 11 m s$^{-1}$ (Cheynet et al., 2024). The smoothed power curve for the 3 MW AWE system does not account for negative power output at the lowest wind speed of 8 m s$^{-1}$ because we assume that the AWE system will not operate at such wind speeds. For the 100 kW AWE system, we averaged two power curves computed at 200 m and 300 m above the surface based on Ranneberg et al. (2018), resulting in a cut-in wind speed of 2 m s$^{-1}$, a cut-out wind speed of 20 m s$^{-1}$ and a rated wind speed of 7.5 m s$^{-1}$. Unlike a wind turbine, the rated power in this curve decreases with increasing wind speed due to power consumption during the retraction phase (Eijkelhof and Schmehl, 2022). Note that because the two power curves for the AWE systems display significantly different rated wind speeds, they lead to large differences in capacity factors, which is discussed in section 4.

In this study, the wind speed values used for the power curve of the AWE systems (Fig. 5) are based on the spatial average of mean wind speeds recorded at altitudes between 200 and 500 m. A similar approach was adopted in Vos et al. (2024) using a

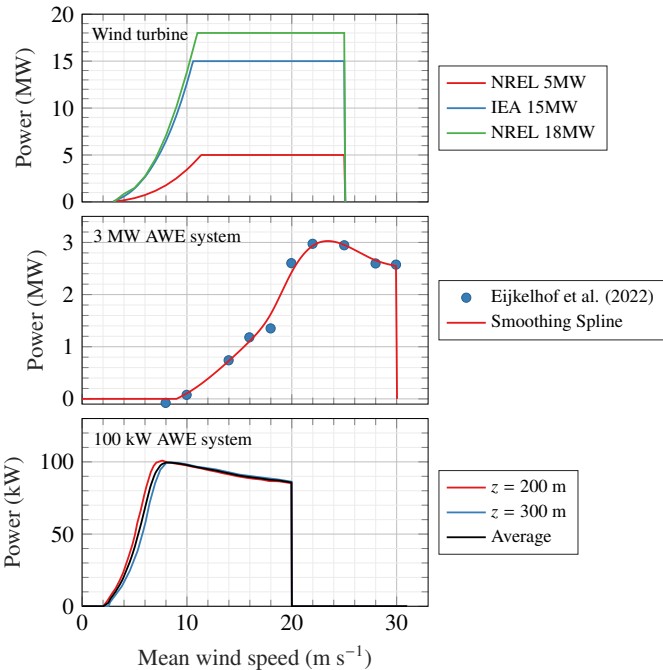

**Figure 5.** Power curves of the wind turbines investigated (top panel), the 3 MW airborne wind energy system presented in Eijkelhof and Schmehl (2022) (middle panel) and power curves for the 100kW AWES from Ranneberg et al. (2018) with averaged operating value between 200 and 300 m (bottom panel).

single height of 350 m above the surface. At the complex site of Bjerkreim, however, the profiler lidar's maximum scanning
height was 300 m, so measurements in this case were limited to the 200–300 m range. The choice of averaged wind speed values
from altitudes between 200 and 500 m serves a dual purpose. Firstly, these altitudes align with the operational heights of large
AWE systems (Eijkelhof and Schmehl, 2022). Secondly, this selection facilitates the approximation of capacity factors for AWE
systems, avoiding the time-consuming task of calculating an optimal flight path for each wind speed profile.

At each site, a reference capacity factor was calculated using lidar data. This reference was then compared to the capacity
factors estimated from the wind atlases, providing a measure of the accuracy of these models in capturing the wind energy
potential. The capacity factor (CF) represents the ratio of the expected output power to the maximum or nominal power output
$P_{\mathrm{max}}$. It can be calculated using time-averaged power output as

$$\mathrm{CF} = \frac{\overline{P(t)}}{P_{\mathrm{max}}} \tag{2}$$

where the overline denotes the temporal average and $P_{\mathrm{max}}$ is the nominal power output. If enough data is collected to construct
reliable probability density functions, the capacity factor can be derived by integrating the product of the wind speed's probability

density function $f_{\mathrm{pdf}}(u)$ and the power curve $P(u)$ across all possible operating wind speeds:

$$\mathrm{CF} = \frac{1}{P_{\max}} \int\limits_{0}^{\infty} f_{\mathrm{pdf}}(u) \cdot P(u) \, \mathrm{d}u \tag{3}$$

In this study, eq. (2) was used for simplicity but also because some of the data collected were recorded over only five weeks, which may be insufficient to construct robust probability density functions as required by eq. (3).

## 4 Results

This section presents the key findings from the comparison between lidar wind speed measurements and model datasets. The analysis includes the five sites, each representing different wind conditions. We first examine the collected wind speed time series and their alignment with model predictions. We then assess the performance of NORA3, ERA5, and NEWA for each site using vertical profiles of error metrics and capacity factor estimates.

### 4.1 Error metrics across sites

Figure 6 presents the time series data collected from lidar measurements and model databases during distinct campaigns at FINO1 (offshore, 2015-2016), FINO3 (offshore, 2013-2014), Bjerkreim (complex terrain, 2010), Sola (coastal terrain, 2013) and Lista (coastal terrain, 2020-2021). Each time series corresponds to wind speed data collected at the range gate closest to 200 m. NEWA does not cover the period of lidar data collection at Lista, limiting the comparative analysis to NORA3 and ERA5. A qualitative analysis of the time series shows that the agreement between the lidar measurements and wind atlases is generally good, with the best match observed at the offshore sites FINO1 and FINO3 but also the coastal site Lista. At the coastal site Sola and the complex terrain Bjerkreim, NORA3 seems to perform better than NEWA and ERA5, partly because it was specifically designed for applications in Northern Europe.

Fig. 7 and Fig. 8 compare four error metrics describing the discrepancies between measurements and modelled mean wind speed data across the five sites at range gates closest to 150 m and 300 m, respectively. These results complement the profiles shown in Fig. 9. Notably, the variability in these metrics across the models observed in Fig. 7 and Fig. 8 aligns with trends at other altitudes, reinforcing the consistency of our findings.

In terms of bias, ERA5 tends to underperform in coastal and complex terrains (Fig. 7 and Fig. 8, panels m and q), while NORA3 consistently shows the lowest bias across all the sites except at FINO1, where it is approximately 0.3 m s$^{-1}$. This differs from earlier studies, such as those by Solbrekke et al. (2021), which reported a smaller bias with a value of 0.14 m s$^{-1}$ at the same site. A similar bias of 0.11 m s$^{-1}$ was obtained by Cheynet et al. (2022) for 2009 alone at 100 m. The positive bias at FINO1 represents an underestimation of the wind speed by the wind atlases. This bias is also found when replacing the lidar data with the cup anemometer data at 100 m above sea level. This larger-than-expected bias may be attributed to the local depletion of wind resources caused by the construction of multiple wind farms around the mast since 2009. This finding is consistent with the study by Podein et al. (2022), which identified an increased wind speed bias between modelled wind speeds and measurements at FINO1 after 2009, coinciding with the start of wind farm development in the area.

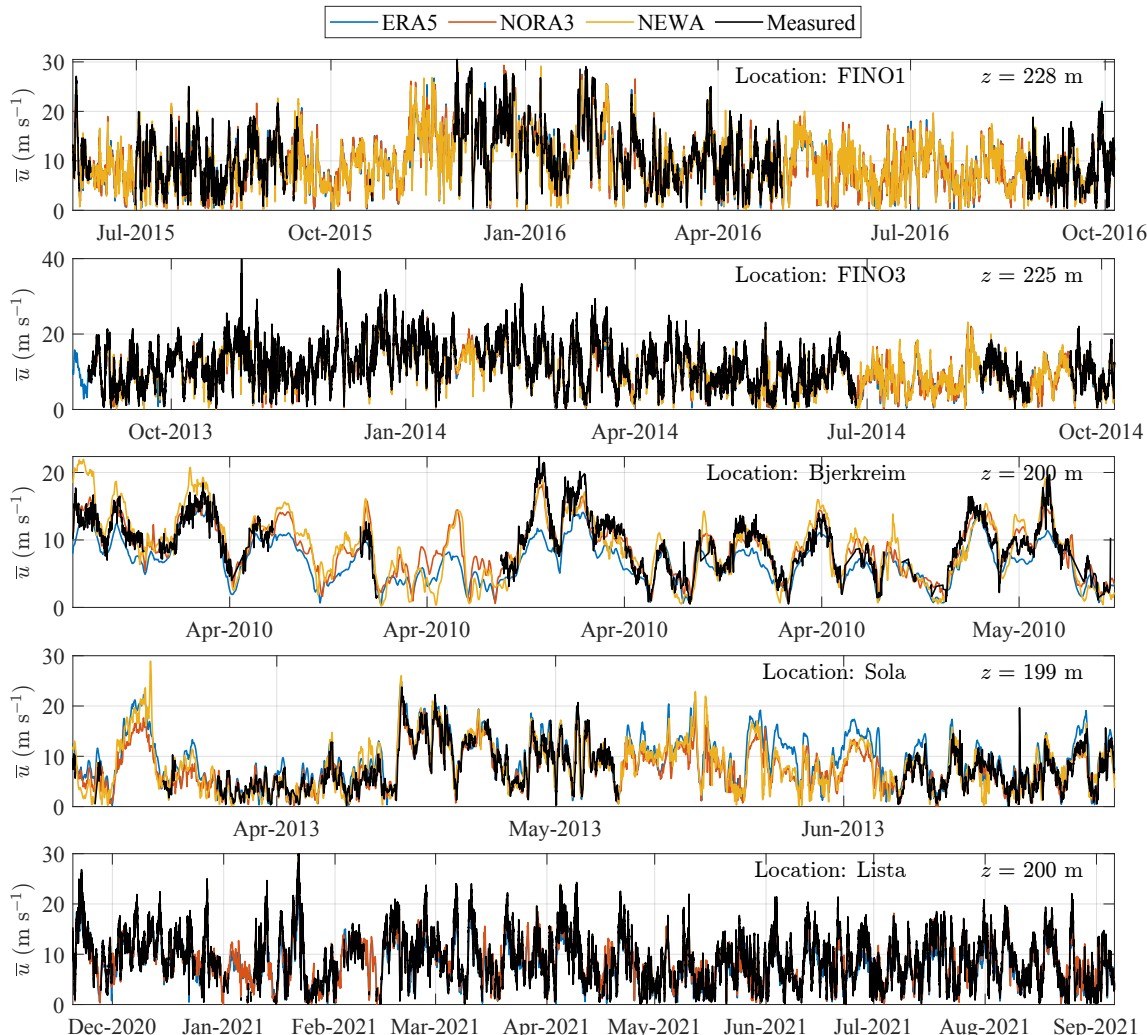

**Figure 6.** Time series of mean wind speed $\overline{u}$ recorded at the five reference sites (FINO1, FINO3, Bjerkreim, Sola and Lista) at the range gate nearest to 200 m. These are superposed with the corresponding time series from the ERA5, NEWA, and NORA3 databases, highlighting periods where lidar data are available.

In panels b and f of Fig. 7 and Fig. 8, NORA3 and ERA5 demonstrate good performance metrics for offshore sites, achieving $R^2$ coefficients close to 0.9, consistent with findings from Cheynet et al. (2022). In complex terrain, NORA3 surpasses NEWA and ERA5, providing the most accurate wind speed estimates as indicated by the highest $R^2$ coefficients, which range from 0.7 to 0.8 for NEWA and ERA5 (Fig. 7 and Fig. 8, panel j). NORA3 also provides, on average, one of the lowest RMSE across different types of terrains. For the two offshore sites, ERA5 and NORA3 perform nearly equally well in terms of RMSE (Fig. 7 and Fig. 8, panels c and g). Thus, NORA3's performance is fairly consistent across diverse topographies, whereas ERA5 may be reliably applied at far offshore sites, particularly where region-specific wind atlases are not available.

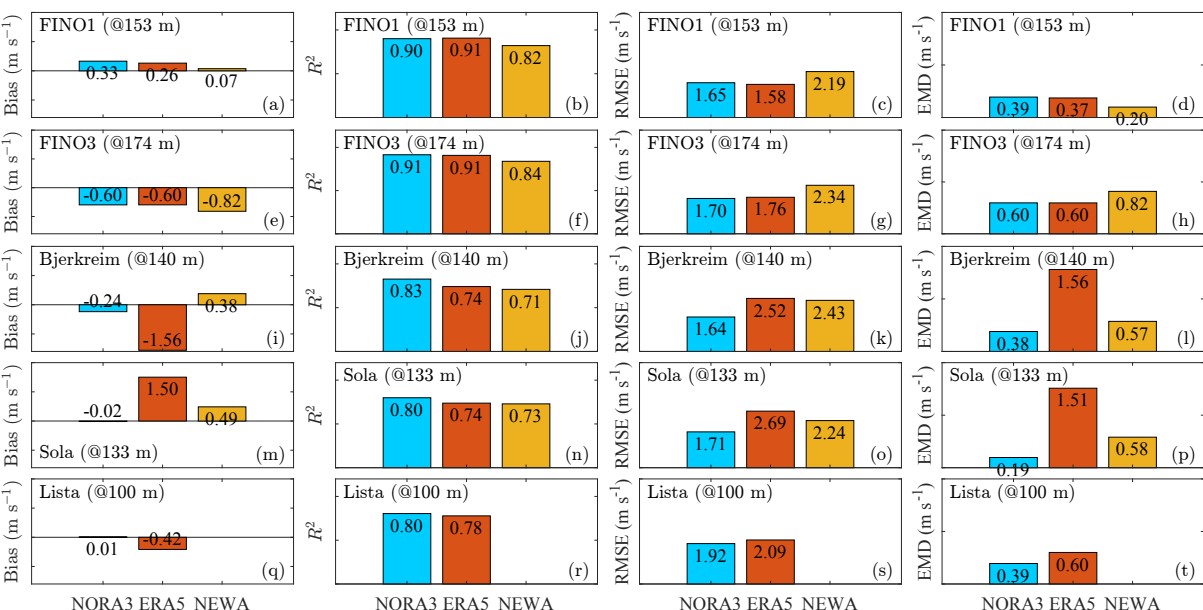

**Figure 7.** Error metrics at the range gate closest to 150 m for measured and modelled mean wind speed data across five sites (FINO1, FINO3, Sola, Bjerkreim, and Lista). Rows represent sites, and columns represent error metrics: Bias, $R^2$, RMSE, and EMD.

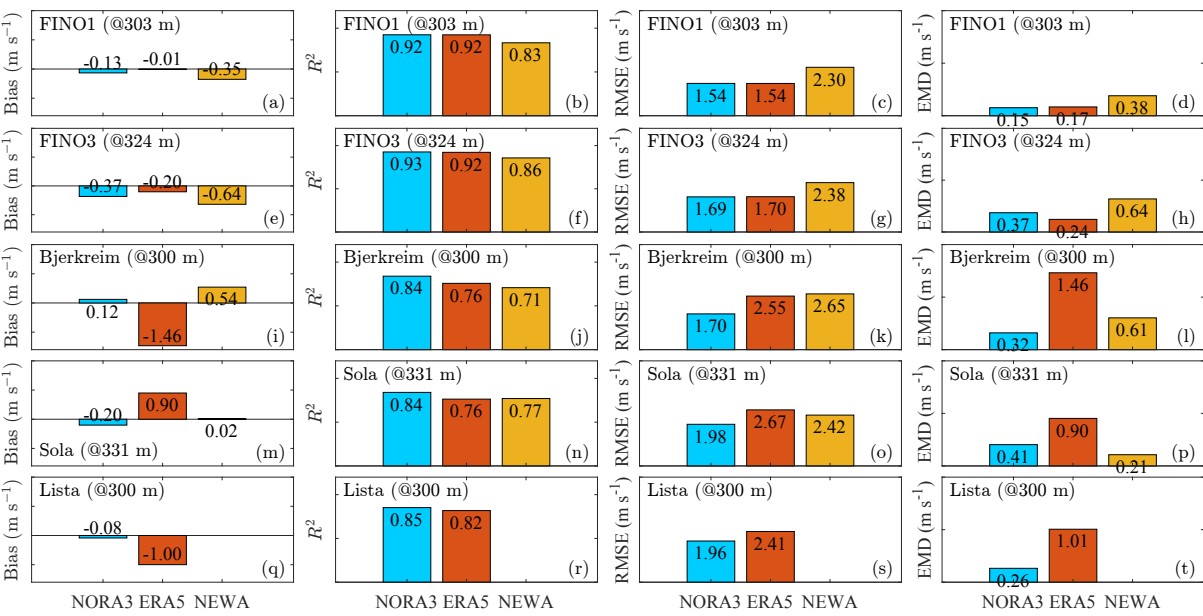

**Figure 8.** Error metrics at the range gate closest to 300 m for measured and modelled mean wind speed data across five sites (FINO1, FINO3, Sola, Bjerkreim, and Lista). Rows represent sites, and columns represent error metrics: Bias, $R^2$, RMSE, and EMD.

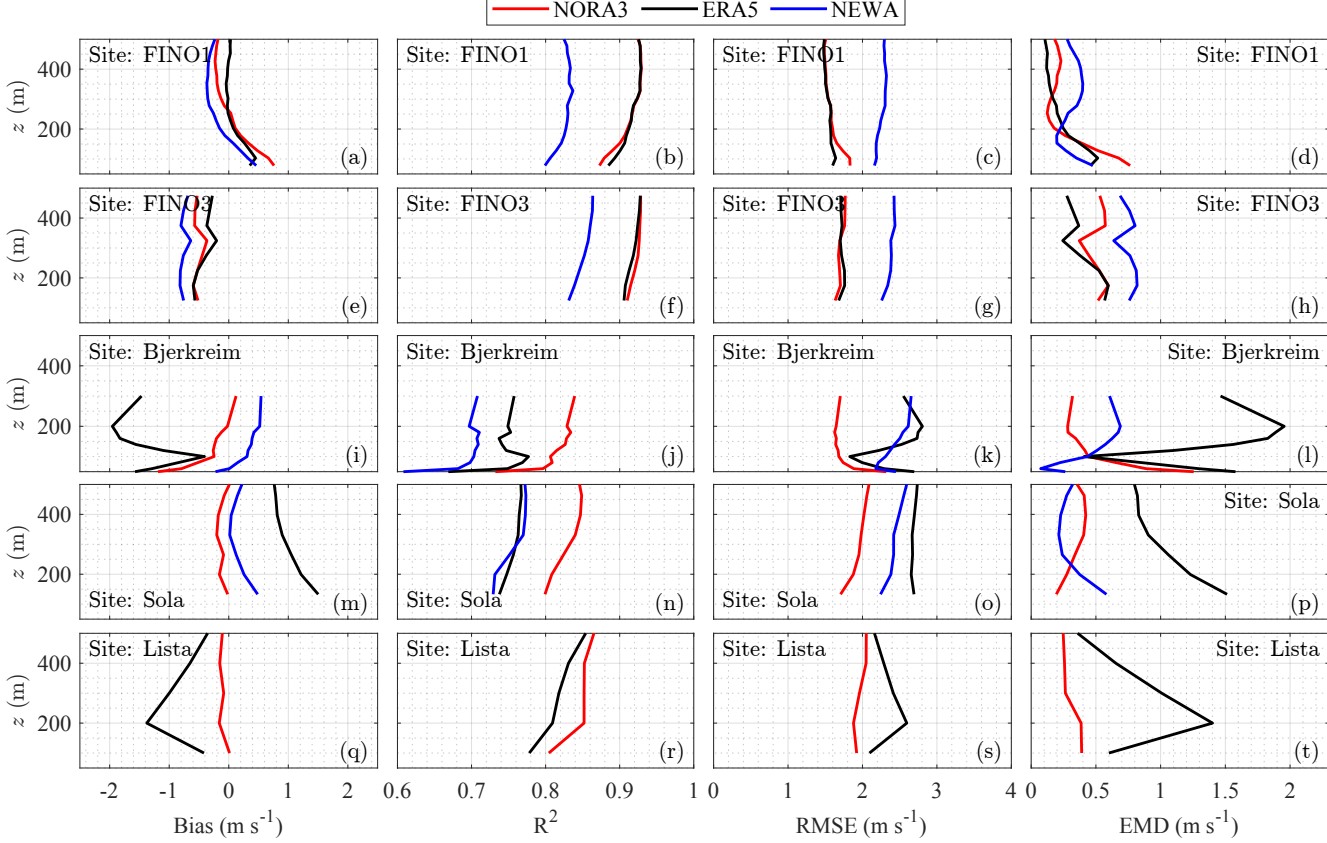

**Figure 9.** Vertical profiles of Bias, $R^2$, RMSE, and EMD of the wind speed across five sites (FINO1, FINO3, Sola, Bjerkreim, and Lista). Rows represent sites, and columns represent error metrics.

The fourth column of Fig. 7 and Fig. 8 displays the EMD. Offshore, NEWA exhibits the lowest EMD at FINO1 and the highest at FINO3 (panels d and h). Although NEWA was specifically engineered with this error metric in mind (Hahmann et al., 2020), the inconsistent EMD values at these two offshore sites may be attributed to the presence of multiple offshore wind farms around FINO1 at the time of data collection. At the coastal sites (Sola and Lista) and complex terrain (Bjerkreim), NORA3 achieves often the lowest EMD, underlining its potential in heterogeneous topographies (Figs. 7 and 8, panels l, p and t). As expected, ERA5 shows significantly higher EMD values than the other two models onshore, which is attributable to its lower horizontal spatial resolution.

The choice of the best model database is not straightforward as it depends on the specific error metrics and the location being analysed (Fig. 7). Furthermore, these error metrics are influenced by the height at which measurements are taken, further challenging the selection of an optimal database for specific applications (Fig. 9).

Figure 9 displays the vertical profiles or Bias, $R^2$ Coefficient, RMSE and EMD of the wind speed for the five sites (FINO1, FINO3, Sola, Bjerkreim and Lista) using the three model databases (NEWA, NORA3 and ERA5) at heights up to 500 m. We

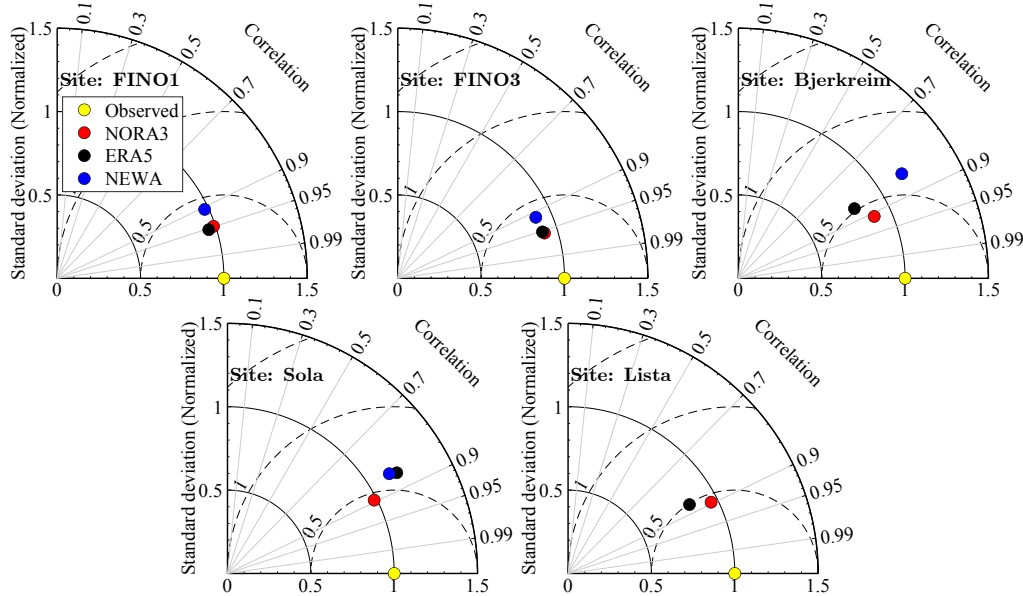

**Figure 10.** Taylor's diagrams for the five sites at the range gate closest to 150 m: FINO1 ($z = 153$ m), FINO3 ($z = 174$ m), Bjerkreim ($z = 140$ m), Sola ($z = 133$ m), and Lista ($z = 100$ m).

remind that the data collection was limited to 300 m at the complex terrain site Bjerkreim and that data from NEWA were not available during the lidar campaign at Lista. In most cases, the agreement between the models and lidar measurements improves with height. However, this trend is not consistently observed at coastal and complex terrain sites, where deviations can occur, depending on the error metric and model database.

In Fig. 9, the bias profiles generally decrease with height at all sites except Bjerkreim, where the results are more nuanced (panel i); NORA3's bias decreases significantly, nearing zero at higher elevations, whereas NEWA's bias increases. In contrast, ERA5's bias is strongly dependent on height. For the $R^2$ and RMSE metrics, NORA3 and ERA5 demonstrate closely matched results at the offshore sites FINO1 and FINO3 across all heights, likely due to NORA3's utilisation of ERA5 inputs as forcing. At the coastal site Sola, NORA3 consistently outperforms NEWA and ERA5 in $R^2$ values at every height. In the complex terrain

of Bjerkreim, NORA3 generally exhibits the lowest RMSE. Meanwhile, NEWA and ERA5 show varying RMSE results based on height, complicating the choice of the most appropriate wind atlases. The analysis of EMD further challenges the selection process. If the EMD is chosen as the preferred error metric, ERA5 emerges as an excellent option offshore, especially for taller wind turbines with hub heights near 150 m. However, in complex terrain such as Bjerkreim, ERA5 performs worst, probably due to its low horizontal spatial resolution. In these areas, NEWA performs best below 100 m, but NORA3 outperforms it at

higher altitudes.

Figure 10 presents Taylor's diagrams, which provide an alternative indicator of the models' performance with respect to the lidar data. These diagrams visualize the standard deviation and correlation coefficient of modelled mean wind speed data compared to measurements at the five sites. In this figure, NORA3 typically shows the best agreement with the measurements,

as its marker is closest to the reference marker representing the observed data, particularly in complex terrain and coastal areas. However, ERA5 and NORA3 perform nearly equally well offshore.

The selection of the most suitable wind atlas depends on the topography, the measurement heights and the desired error metrics. ERA5 can be considered a versatile choice for offshore wind energy applications, extending beyond European waters. NEWA may be appropriate when focusing on specific error metrics like EMD, and NORA3 may deliver consistently high performance at the Norwegian onshore and offshore sites. It should also be noted that the variability in model performance across the sites may partly be attributed to the different lidar instruments. The DWL profiler deployed in Bjerkreim in 2010 was a now-discontinued WindCube V1; a scanning lidar WindCube 100S was used at the coastal site Sola and the FINO1 platform, and a discontinued WindCube WLS70 was deployed at the offshore site FINO3 and the coastal site Lista. Furthermore, each lidar's performance is inherently unique due to the fine-tuning of the hardware during the manufacturing and calibration process, the discussion of which is beyond the scope of this study. The discrepancy between the modelled wind speed data and lidar-based measurements in the complex site Bjerkreim or the coastal sites Sola and Lista is also influenced by the higher occurrence of non-homogeneous flow fields at onshore sites compared to offshore, particularly within the first 300 m above the surface, which can exacerbate the measurement uncertainties of lidar retrievals using DBS or velocity-azimuth display scanning modes (Klaas-Witt and Emeis, 2022).

## 4.2 Capacity factor estimates

This section analyses the CFs of the turbine models and AWE systems (section 3.3) at the five sites. The estimates are derived from time series of wind speed measured by lidars and provided by the three wind atlases. Hereinafter, wind speeds from both the lidar instruments and the wind atlases are interpolated to the hub height of the wind turbines or spatially averaged over the operational height of the airborne wind energy (AWE) systems, ranging from 200 to 300 m above the surface (Fig. 5).

Each row of Fig. 11 represents a specific location, while each column refers to a different turbine type. For the coastal and complex terrain sites, we calculated capacity factors for turbines with larger nameplate capacities than typical onshore models. Also, the CFs presented are not indicative of the sites' climatology, as the measurement campaigns were significantly shorter than the standard 30-year period. Thus, the CF should serve solely for comparison between the model and the measurement data, and not for evaluating the wind energy potential at these sites.

The offshore site FINO3 (Fig. 11, panels d, e and f) demonstrates the best agreement between model datasets and lidar measurements in terms of CF. However, the NREL 5 MW and the IEA 15 MW wind turbines exhibit substantial differences in CF, largely due to variations in hub heights. For the 15 MW wind turbines at FINO3, the estimated CFs differ by a maximum of 0.03 between measurements and ERA5. At FINO1 (Fig. 11, panels a, b and c), the discrepancies between the model and measurements are greater than at FINO3, which might be due to the presence of wind farms around the lidar.

At the coastal sites Sola and Lista (Fig. 11, panels j-o), the NORA3 database provides nearly identical CFs to the lidar data. At Sola, ERA5 significantly overestimates the CF by about 40% for the NREL 5 MW wind turbine and 38% for the IEA 15 MW wind turbine. In contrast, ERA5 slightly underestimates the CF of all turbine models at Lista. The NEWA hindcast also

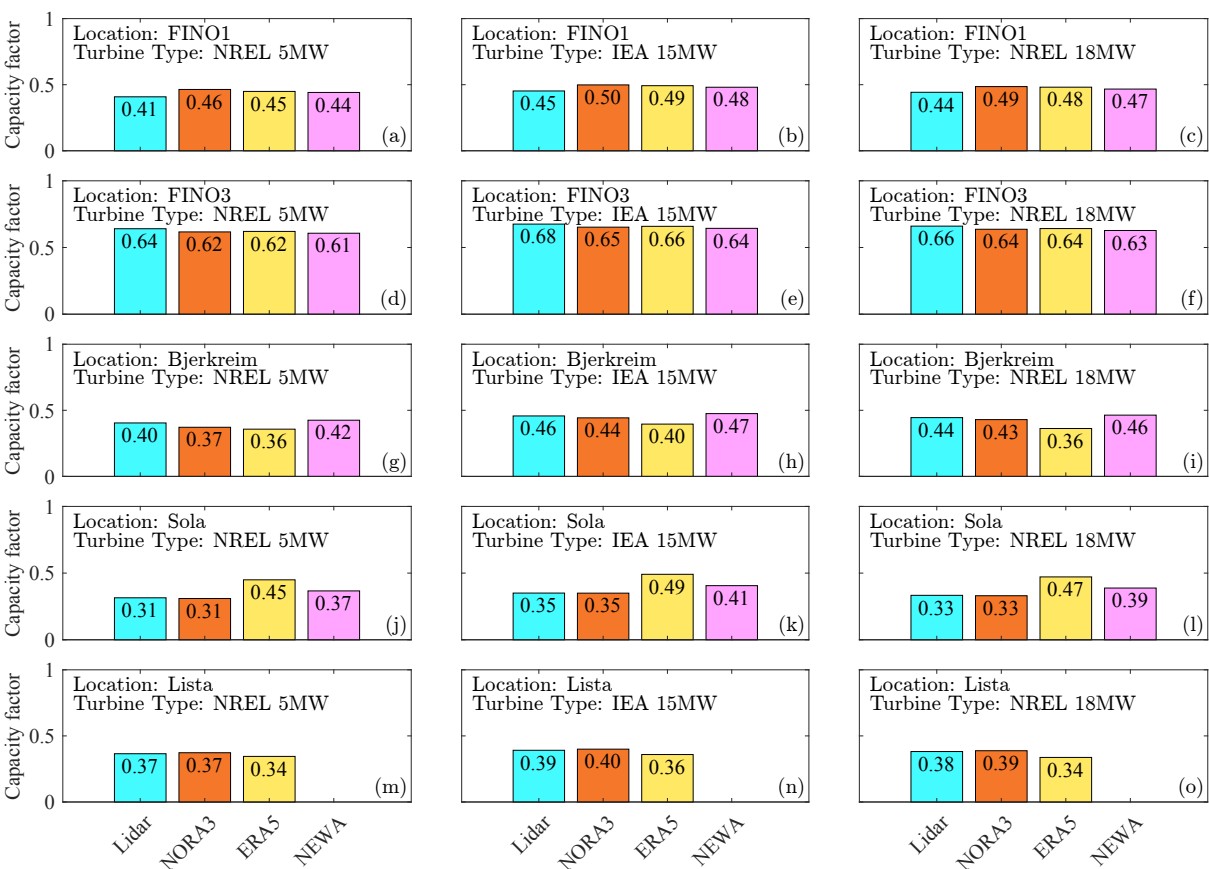

**Figure 11.** Estimated capacity factors for different turbine models (NREL 5 MW, IEA 15 MW and NREL 18 MW) at the five sites, using lidar and three model datasets as wind inputs.

overestimates the CF at Sola, but to a lesser extent: about 16% for the NREL 5 MW turbine and 14% for the IEA 15 MW turbine.

At the complex terrain site Bjerkreim (Fig. 11, panels g, h and i), the NORA3 and ERA5 datasets slightly underestimate the
405 CFs for the NREL 5 MW wind turbine. However, NEWA provides CFs that are closest to those measured by lidar. This finding is supported by Fig. 9, which indicates that the EMD, a good metric for wind resource assessment, is lower for NEWA than that for ERA5 and NORA3 below 100 m at Bjerkreim.

Figure 12 shows the CFs for the 3 MW and 100 kW AWE systems at the five sites of interest. Offshore, the CFs range from 0.13 to 0.22 for the 3 MW system and from 0.66 to 0.71 for the 100 kW system. This significant difference arises from the
410 100 kW system's lower cut-in (2 m s$^{-1}$) and rated (7 m s$^{-1}$) wind speeds compared to the 3 MW system's cut-in (9 m s$^{-1}$) and rated (22 m s$^{-1}$) wind speeds. Trevisi et al. (2021) used a less conservative power curve for a 3 MW AWE system with a cut-in wind speed of 2 m s$^{-1}$ and a rated wind speed between 7 and 8 m s$^{-1}$, which led to a CF of 0.64. Although the 3 MW system used in our study displays a higher cut-out wind speed (30 m s$^{-1}$) than the 100 kW system, it does not sufficiently compensate

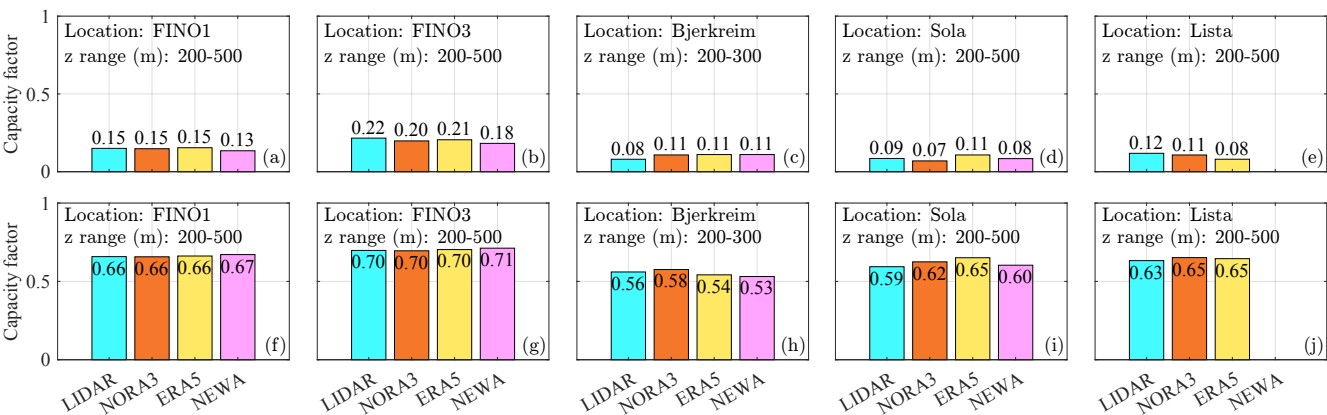

**Figure 12.** Estimated capacity factors for the 3 MW (top panels) and the 100 kW (bottom panels) AWE system at the five sites, using lidar and three wind atlases as wind inputs.

for the lower CF. Therefore, the economic success of larger AWE systems may depend on incorporating lower cut-in and rated wind speeds.

At the offshore sites FINO1 and FINO3 (Fig. 12, panels a, b, f and g), the CFs of AWE systems derived from wind atlases and lidar datasets show similar values, whereas larger differences emerge when using wind turbine data (Fig. 11). The small CF discrepancies between simulated and lidar data at FINO1 for AWE systems suggest that measurements at higher altitudes effectively reduce the influence of wind farm wake on wind flow. Figure 12 implies also that discrepancies between measured and modelled wind speed in terms of CF may decrease with increasing altitude.

For the coastal sites Sola and Lista (Fig. 12, panels d, e, i and j) and the complex terrain site Bjerkreim (Fig. 12, panels c and h), the discrepancies in CF for the 3 MW AWE system are also minor. This is likely because wind speeds below 9 m s$^{-1}$, which fall below the system's cut-in speed, are not included in the CF calculation. This avoids erroneous flow modelling at moderate and low wind speeds, which sometimes reflect strongly stable atmospheric boundary layers that are not always well captured by wind atlases (Holtslag et al., 2013). Deviations between models and measurements are also documented at wind speeds above 20 m s$^{-1}$, either near the surface (Bentamy et al., 2021; Gandoin and Garza, 2024) or within the first 100 m above the surface (Solbrekke et al., 2021). Nevertheless, such deviations may have a limited impact on the CF of wind energy systems with a relatively low cut-out speed of around 20 to 25 m s$^{-1}$.

Although this subsection primarily examines the CF, which is tied to the levelized cost of energy, it represents just one of several metrics used to assess the performance of intermittent renewable energy systems (Simpson et al., 2020). Alternative metrics, such as the Gini coefficient (Malz et al., 2020) and the correlation coefficient between different renewable energy sources such as solar and wind (Malz et al., 2020; Jurasz et al., 2020), also provide valuable insights into the variability of power output in AWE systems and wind turbines. However, examining the complementarity of AWE systems with other energy sources, such as wind and solar, falls outside the scope of this study.

## 5   Discussions

This section explores the challenges and opportunities for improving wind energy assessment for airborne wind energy systems and large wind turbines with a nameplate capacity of 15 MW or more. We first address the need for more capable wind profilers to meet the demands of larger turbines and AWE systems. We then discuss the limitations of mesoscale models in complex and coastal terrains and the potential benefits of combining them with microscale models. These considerations suggest some pathways for improving wind flow simulations and instrument design in future research.

### 5.1   A need for more powerful wind profilers?

When commercial DWL profilers became available in the 2000s, wind turbines had a nominal capacity of about 5 MW, with tip heights around 150 m. This made the typical profiler lidar scanning range of 200–300 m sufficient at the time. However, wind turbine sizes have grown significantly since then. For instance, in 2024, Mingyang Smart Energy installed a 20 MW turbine with a tip height potentially reaching 300 m (Casey, 2024). This growth, coupled with the rising interest in AWE systems operating at heights between 200 and 600 m, underlines the need for a new generation of DWL instruments capable of reliably profiling winds up to 500 m above the surface or even higher. While scanning lidars can extend profiling range, they are often heavier, more expensive, and less reliable, as they contain more moving parts and are not specifically designed for wind profiling. Profiler lidars with a range exceeding 300 m remain uncommon among commercial lidar producers. One notable exception is Halo Photonics by Lumibird, which has developed the BEAM 6X series, capable of measuring wind speeds up to 500 m (Halo Photonics, 2024). To date, however, we could not find any studies that demonstrate the validity of measurements from this lidar.

### 5.2   Mesoscale limitations and microscale needs

Wind simulations for wind energy applications are typically performed using two types of models: mesoscale models, which provide wind speed data over spatial scales ranging from a few kilometres to hundreds of kilometres, and microscale models, which operate at smaller scales, from a few metres to a few kilometres. While these models are complementary, microscale models are particularly useful in capturing wind flow variability in complex terrain, where topographic features, infrastructure or wind farms significantly influence wind conditions.

This study primarily focuses on mesoscale-derived wind speed data, which can be limited in capturing fine-scale flow features in complex terrains or near coastal sites. For offshore sites like FINO3, microscale effects are likely negligible. However, for coastal sites such as Sola and Lista, and the complex terrain at Bjerkreim, microscale modelling may improve the agreement between simulated and measured wind speeds. At Bjerkreim, computational fluid dynamics (CFD) models could help capture complex phenomena, such as flow recirculation and detached downslope flow, which are prevalent in mountainous terrain like Southeastern Norway. At FINO1, microscale flow simulations may also be needed to model wake effects on wind speed measurements. Future studies should investigate the benefits of coupling mesoscale and microscale models to enhance performance metrics at coastal and complex sites. The comparison conducted in this study remains valuable because microscale

models, while potentially more precise than mesoscale models in complex terrain, depend on accurate initial and boundary conditions that can be provided by the mesoscale models.

## 6  Conclusions

This study examines the capability of three wind atlases, NORA3, NEWA, and ERA5, in modelling wind speed profiles up to 500 m above the surface for wind energy applications. Reference wind speed profiles were obtained from Doppler wind lidar (DWL) measurements conducted at five distinct sites in Northern Europe. These sites encompass diverse topographies such as flat coastal terrain, mountainous regions, and offshore environments. The study aims to broaden the validation scope to altitudes critical for large wind turbines and airborne wind energy (AWE) systems. This study addresses a significant challenge in wind energy, as there has been relatively limited investigation into tall wind speed profiles using scanning DWL in profiler modes for wind resource assessment at heights up to 500 m above the surface.

The study found that the three wind atlases perform well in offshore locations, with ERA5 and NORA3 showing the closest correlation to lidar data. More specifically, NORA3 and ERA5 perform almost equally well in terms of correlation coefficient, root mean square error (RMSE) and earth mover's distance (EMD). However, ERA5 has a lower bias above 200 m. Onshore, NORA3 outperforms ERA5 and NEWA at all heights for most error metrics. While the agreement between the models and lidar measurements generally improves with height, this trend is less consistent at coastal and complex terrain sites, where significant deviations occur, especially for ERA5 and NEWA.

In terms of wind turbine capacity factor (CF), all datasets show good agreement with CF derived from lidar data offshore, particularly for the largest turbines. However, at the FINO1 site, the three models overestimate the CF, likely due to local wind resource depletion from surrounding offshore wind farms. In coastal terrain, NORA3 provides excellent CF agreement with lidar data, NEWA performs reasonably well, but ERA5 overestimates the CF. In complex terrain, NEWA and NORA3 both perform well, while ERA5 substantially underestimates the CF.

For AWE systems, the CF was fairly consistent across all the wind atlases but showed considerable dependency on the AWE design. Smaller AWE systems with lower cut-in and rated wind speeds achieved higher CFs, whereas the larger 3 MW AWE system considered in this study was penalised by its high cut-in and rated wind speeds. Therefore, designing larger AWE systems with lower cut-in wind speeds is essential to reduce their levelized cost of energy. Finally, the development of DWL technology must keep pace with the growing size of wind turbines, requiring more powerful profilers to avoid the increased costs associated with the deployment of scanning lidar instruments for tall wind profiles.

This study was based on relatively limited datasets: the temporal coverage is insufficient to represent a full climatology timescale and the spatial coverage is restricted to a few locations in Norway and the North Sea. Although the sites selected in this study provide a diverse range of topographies, having more locations from additional countries would enhance the robustness of the findings. Additionally, the wind atlases have a limited temporal resolution of 30 to 60 minutes, which may not adequately capture short-term variations in wind speed profiles. Finally, it should be noted that wind speed profiles established by DWLs are typically validated against anemometers mounted on met masts. However, such comparisons become impractical at altitudes

above 200 m. Consequently, the accuracy of wind speed profiles at these heights, as measured by profiler lidar or scanning lidar in profiler mode, requires further evaluation.

The general conclusion is that NORA3 excels onshore, while ERA5, with its global coverage, performs equally well offshore. Onshore data quality is slightly lower for all datasets due to the complexity of wind patterns over land. In particular, ERA5 shows significant height-dependent errors, possibly due to its lower spatial resolution compared to NEWA and NORA3. These findings highlight the importance of selecting the appropriate wind atlas and error metrics to improve wind resource assessment. This selection should be tailored to the specific wind energy system, site, and operating height.

## Appendix A: Error metrics with non-linear wind speed regression

A non-linear regression was also tested as an alternative to interpolation for smoothing the vertical wind speed profiles up to 500 m for NEWA, NORA3 and ERA5. The regression relies on fitting a modification of the wind profile model by Deaves and Harris (1982). This modified analytical function combines a classic logarithmic profile with a third-order polynomial function and is expressed as

$$\overline{u}(z) = \frac{u_*}{\kappa} \ln\left(\frac{z}{z_0}\right) + p(z), \text{ with} \tag{A1}$$

$$p(z) = a_1 z + a_2 z^2 + a_3 z^3, \tag{A2}$$

where $z$ is the height above the surface; $\kappa \approx 0.4$ is the von Kármán constant and $z_0$ is the roughness length. The coefficients $a_i$, where $i = \{1, 2, 3\}$, are determined empirically by least-squares fit (Cheynet et al., 2024). For the coastal and complex terrain sites, the roughness length is approximated by the values 0.01 m and 0.1 m, respectively, following the traditional roughness length classification onshore (Wieringa, 1980, 1986). These values are realistic enough to ensure a reasonable fit in the lower part of the ABL. Above the ocean, the roughness length is estimated using Charnock's relationship (Charnock, 1955), which quantifies the dependency of the roughness length on the sea state:

$$z_0 = \frac{a}{g} u_*^2 \tag{A3}$$

where $g = 9.81 \text{ m s}^{-2}$ is the gravitational acceleration, and $a \approx 0.014$ is an empirical coefficient (Kraus and Businger, 1994). Equation (A3) is combined with the neutral logarithmic wind speed profile (Kaimal, 1994)

$$\overline{u}(z) = \frac{u_*}{\kappa} \ln\left(\frac{z}{z_0}\right) \tag{A4}$$

into a new equation:

$$z_0 - \frac{a}{g} \left[\frac{\kappa \overline{u}(z_r)}{\ln(z_r/z_0)}\right]^2 = 0, \tag{A5}$$

which is solved for $z_r = 10$ m and provides an estimate of the roughness length in the marine ABL.

Figure A1 and Fig. A2 present the wind speed error metrics used to evaluate the performance of the wind atlases against lidar measurements, both at the range gate nearest to 150 m and across multiple heights up to 500 m at the five investigated

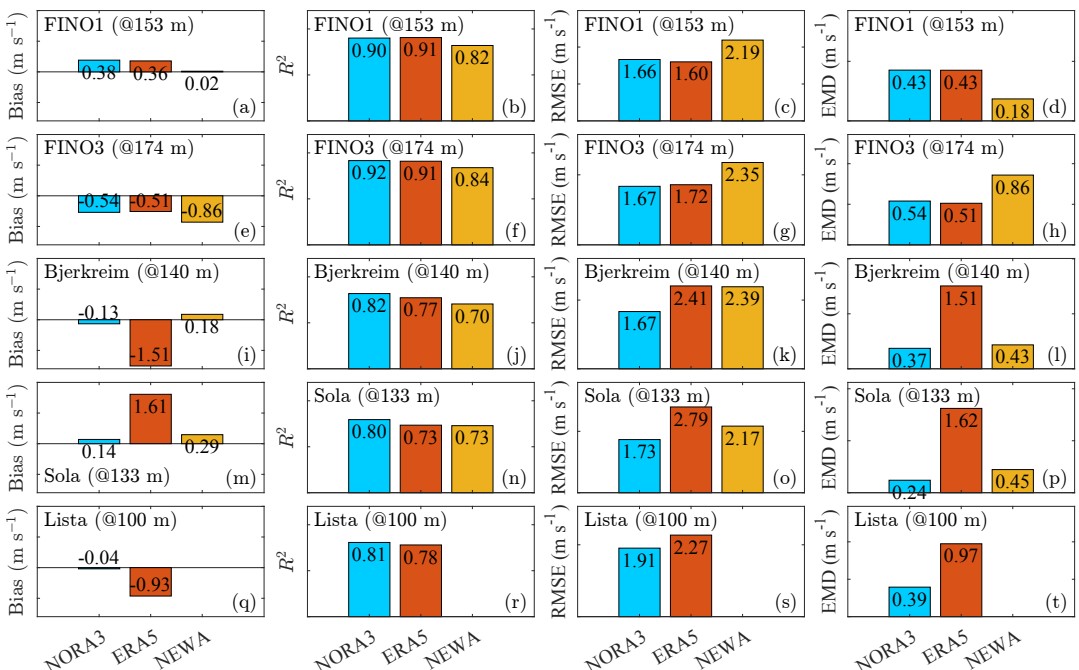

**Figure A1.** Error metrics at the range gate closest to 150 m for measured and modelled mean wind speeds across five sites (FINO1, FINO3, Sola, Bjerkreim, and Lista). Rows represent sites, and columns represent error metrics: Bias, $R^2$, RMSE, and EMD. Wind speed profiles were aligned using non-linear regression instead of linear interpolation.

sites. The non-linear regression produces smooth profiles of error metrics but does not necessarily minimise the error metrics themselves. Several thousand samples were used to compute these ensemble-averaged error metrics, which help smooth out
potential discrepancies caused by linear interpolation. Consequently, when sufficiently large datasets are available, the linear interpolation of vertical wind speed profiles at additional height levels yields reliable results and was identified as the most robust approach in this study.

*Data availability.* The NEWA data were collected from the "New European Wind Atlas", which is a free, web-based application developed, owned and operated by the NEWA Consortium. For additional information see www.neweuropeanwindatlas.eu. The ERA5 reanalysis
data used in this study are publicly available through the Copernicus Climate Change Service (C3S) Climate Data Store (CDS) at https://cds.climate.copernicus.eu/. The NORA3 reanalysis data used in this study are publicly available through the Norwegian Meteorological Institute's THREDDS server at https://thredds.met.no/thredds/projects/nora3.html. The modified datasets used to generate the figures in this study will be made publicly available on Zenodo under a BSD-3 open-access license.

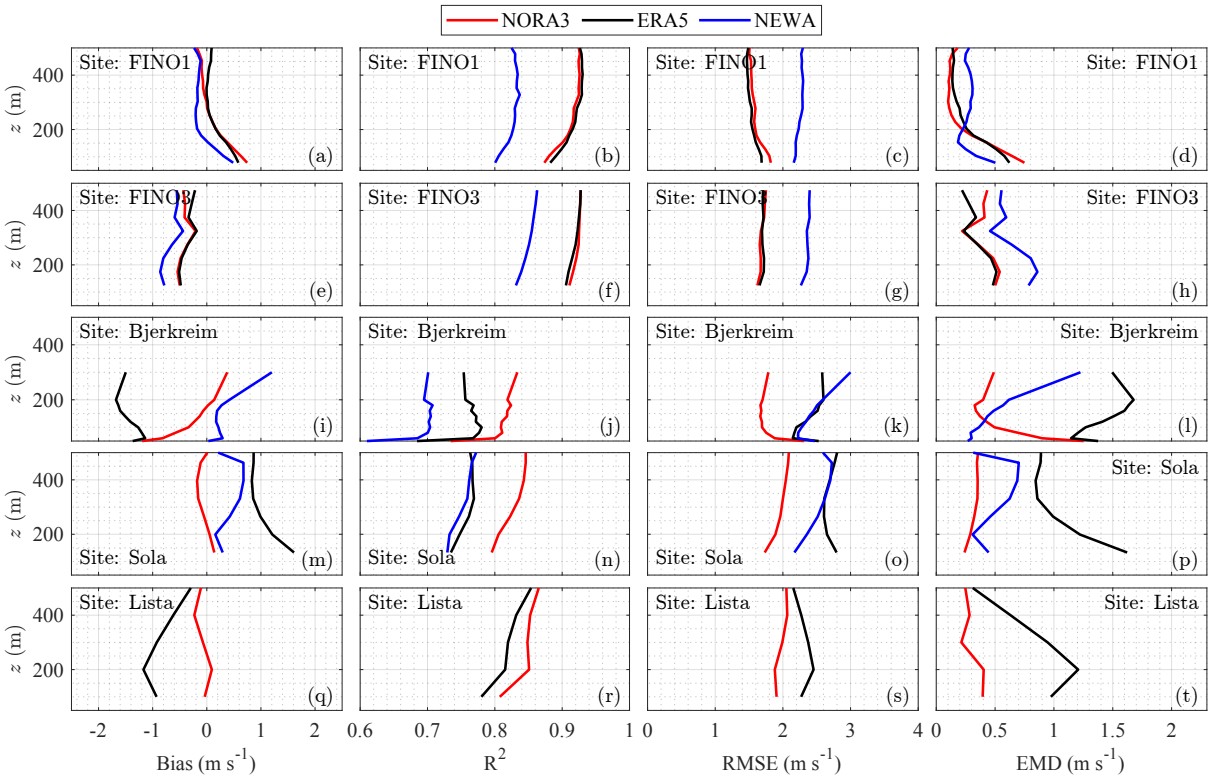

**Figure A2.** Vertical profiles of Bias, $R^2$ Coefficient, RMSE, and EMD of horizontal wind speed based on non-linear regression across the five sites: FINO1, FINO3, Sola, Bjerkreim and Lista. Each row represents one site and each column represents one error metric.

*Author contributions.* JR, JMD, HH and ØB contributed to the conceptualization of the study. EC, JMD, and JR developed the methodology. 540 Data collection was carried out by EC, JMD, JR, and AP, while data analysis was performed by EC and JMD. The original draft was written by EC and JMD, and all authors participated in the review and editing process. Supervision was provided by JR and EC, and JR was responsible for funding acquisition.

*Competing interests.* Etienne Cheynet and Alfredo Peña are members of the editorial board of Wind Energy Science.

*Acknowledgements.* This work is co-funded by the European Union's Horizon 2020 research and innovation program under the Marie 545 Sklodowska-Curie grant agreement No. 858358 (LIKE –Lidar Knowledge Europe, H2020-MSCA-ITN-2019), the projects Large Offshore Wind Turbines (LOWT) (project number: 325294) and ImpactWind Southwest (project number: 332034) funded by the Research Council of Norway and the Marie Sklodowska-Curie grant agreement No. 101168855 (NEXTgenT - NEXT generation of over 25MW offshore wind Turbine rotor design, HORIZON-MSCA-2023-DN-01).

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
