# Peer review of "Tall wind profile validation of ERA5, NORA3 and NEWA, using lidar observations"

_Wind Energy Science, 2024_

## Referee Comment (RC1)

Review report "Tall Wind Profile Validation Using Lidar Observations and Hindcast Data", Cheynet et al.

In this manuscript three model-based datasets (NORA3, NEWA and ERA5) are validated using Doppler wind lidar data obtained from five locations, including North Sea (FINO1, FINO3) and coastal and complex terrain locations in Norway. Emphasis is given to long-range scanning Doppler wind lidars, providing wind profiles far above the atmospheric surface layer that are relevant for modern wind turbine designs and airborne wind energy (AWE) systems. These altitudes that are not feasible for in-situ wind measurements in tall masts (especially offshore) or the more extensively used short-range Doppler lidar wind profilers that are limited to 200-300m. The validation focuses on altitudes between 100m and 500m, using various error metrics, and their corresponding capacity factors, based on power curves for various wind turbines and AWE systems. The authors claim an increasing agreement between the models and the measurements with height, and argue that those models are valuable R&D on AWE systems.

In general, the manuscript addresses an important point, namely the need to validate models at altitudes relevant for future wind energy systems, and the lack of continuous, long-term measurement campaigns to do so. The authors point to the need of dedicated Doppler wind lidar profilers with sufficient height range, which are indeed lacking commercially right now. The manuscript also highlights the complexity of comparing the performance of various models, and that the best choice really depends on their actual application (type of location, relevant height range, …).

**General comments**

1. I have some objection to the term "tall wind profile". Tall is used for physical structures, like masts or wind turbines, but a wind profile cannot be tall. I am not aware that "tall wind profile" is a commonly used term in our community, however, if I am wrong in this (i.e. it is used in more papers), I will drop my objection.

2. The FINO1 measurements are not suitable for validation due to the presence of an operating wind farm (and the models do not include that). Therefore it should be not be included here, also because FINO3 is already available to cover the offshore situation. Only if the data could be filtered to minimize the effect of the wind farm (for instance, if the influence is only present for certain wind directions), its inclusion would make sense.

3. The authors note that the conclusions on the model performances for the different sites might be hampered by the quality of the different Doppler lidar instruments. However, those measurements have been validated with other measurements, as described in Section 2.2. Wouldn't it therefore not be possible to quantity whether the validation results are significant in terms of the measurement uncertainty or bias?

4. For the wind measurements at the relevant altitudes the authors immediately jump to Doppler lidar in the introduction. But there also other remote sensing instrument that can measure wind between 200 and 500m, like radar wind profilers and sodar. In fact, for this validation study, their temporal and vertical resolution would be more than sufficient. The choice of Doppler lidar should be given a bit more context and motivation.

5. The conclusion that there is an increasing agreement between models and lidar measurements, as stated in the abstract, is not explicitly stated in the main text, including the conclusion. Either the main text is underselling the results, or the abstract is overselling it.

6. I was a bit surprised that although the paper emphasis the need for wind profile beyond what can be reached by traditional masts and (floating) short-range wind lidars, still most of the presented results are at an altitude of 150m (for which, by the way, there are much more lidar data available, including offshore). Why this particular choice?

**Specific comments**

A. Title: "validation using lidar observations and hindcast data". Are you not validating hindcast data using lidar observations?

B. Table 1, why this table is in the manuscript? To make the point that there are very limited amount of tall towers with in-situ wind measurements, such a table is not required.

C. Section 3.3: It is not explained how the wind profile is used in the calculation of CF for wind turbines. Is this wind speed at hub height taken or a rotor average. Table 4 provides hub height and rotor diameter of the various wind turbine types, but nowhere it written how this information is used. This is in contrast to the extension discussion on the AWE system.

D. In correct usage of term "In-situ" throughout the manuscript. Doppler lidar is a remote-sensing instrument and definitely not "in-situ"! However, in distinguishing between model and measurement data, in several parts of the manuscript the term "in-situ" is used for Doppler lidar, which is wrong. This needs to be corrected.

E. The distinction coastal and complex locations from Figure 2 is not clear (at least for the non-Norwegian reader). Would a zoom-in of the map help to clarify the difference between the Sola and Lista as coastal/non-complex, and Bjerkeim as complex terrain?

F. Section 5.2: Could you be more explicit, or give examples, on what you mean with "microscale models".

G. Section 5.2: At the end of this section the issue of Doppler lidar wind profiling measurements in complex terrain is mentioned. This is a relevant point, but doesn't belong to this section (which is about the models). Maybe this issue should be discussed much earlier in the paper. Are there solutions to this issue, or would validation in complex terrain remain problematic?

---

## Referee Comment (RC2)

Review of "Tall Wind Profile Validation Using Lidar Observations and Hindcast Data"

**General comments**

This paper addresses an important and timely topic by validating three widely used wind reanalysis and hindcast models—NORA3, NEWA, and ERA5—against lidar measurements at five strategic locations in the North Sea and along the Norwegian coast. The validation focuses on wind speed profiles at heights relevant to modern wind turbines and emerging airborne wind energy systems (100-500 m), making this study directly applicable to the future of wind energy technology.

The study effectively uses appropriate error metrics, including the Earth Mover's Distance (EMD), to evaluate model performance across offshore, coastal, and complex terrain sites. The findings emphasize the critical need to select appropriate wind atlases based on site-specific geography and altitude, particularly in complex terrain where regional models like NORA3 tend to outperform global datasets like ERA5. The study also underscores the need for more tailored lidar wind profilers to accommodate the growing size of modern wind turbines and the emerging technology of airborne wind energy systems.

While the paper provides valuable insights, it acknowledges limitations in the temporal scope, as the datasets do not cover a full climatology period. The authors suggest expanding measurement sites and improving temporal resolution in future studies to strengthen conclusions. Overall, this study makes a significant contribution to the ongoing effort of properly validating reanalysis models for the evolving wind energy sector.

**Specific comments**

1. Why is the FINO1 platform used for model validation when it is located near several wind farms? As noted in the manuscript, this proximity likely affects the measurements, making FINO1 unsuitable for validation unless the models explicitly account for the wind farms or the data are filtered to exclude disturbed wind directions. Since the measurements at FINO3 do not have nearby wind farms, wouldn't they already provide a more representative view of undisturbed offshore conditions?

2. In line 300, it is mentioned that the EMD values are comparable across all models at coastal locations. However, this is not the case for the Sola site, where there are noticeable differences between the models.

3. The paper emphasizes the validation of hindcast data at higher altitudes, beyond what has been extensively studied. Given this, why focus on results at 150 m, a height already typical for current wind turbines, when higher-altitude data are available? The higher-altitude comparisons would seem more aligned with the study's stated objectives.

**Technical corrections**

1. In the introduction, it might be appropriate to add the reference, where they use ERA5 to compute AEP of airborne wind energy systems:
   *Schelbergen, M., Kalverla, P. C., Schmehl, R., and Watson, S. J.: Clustering wind profile shapes to estimate airborne wind energy production, Wind Energy Science, 5, 1097‑1120, https://doi.org/10.5194/wes-5-1097-2020, 2020.*

2. In line 54, the acronym "AWE" is repeated unnecessarily. Please use the acronym directly after the first mention.

3. In line 111, it is generally not proper styling to add directly an url to the text. Please include it in the references and refer to that.

---

## Author Comment (AC1)

Responses to Reviewers' Comments for Manuscript
WES-2024-119

**Tall Wind Profile Validation Using Lidar Observations and Hindcast Data**

Addressed Comments for Publication to

Wind Energy Science (ISSN 2045-2322)

by

Cheynet et al.

**1 Authors' Response to Reviewer 2**

**1.1 General comment**

> **General Comments.** This paper addresses an important and timely topic by validating three widely used wind reanalysis and hindcast models—NORA3, NEWA, and ERA5—against lidar measurements at five strategic locations in the North Sea and along the Norwegian coast. The validation focuses on wind speed profiles at heights relevant to modern wind turbines and emerging airborne wind energy systems (100-500 m), making this study directly applicable to the future of wind energy technology. The study effectively uses appropriate error metrics, including the Earth Mover's Distance (EMD), to evaluate model performance across offshore, coastal, and complex terrain sites. The findings emphasize the critical need to select appropriate wind atlases based on site-specific geography and altitude, particularly in complex terrain where regional models like NORA3 tend to outperform global datasets like ERA5. The study also underscores the need for more tailored lidar wind profilers to accommodate the growing size of modern wind turbines and the emerging technology of airborne wind energy systems. While the paper provides valuable insights, it acknowledges limitations in the temporal scope, as the datasets do not cover a full climatology period. The authors suggest expanding measurement sites and improving temporal resolution in future studies to strengthen conclusions. Overall, this study makes a significant contribution to the ongoing effort of properly validating reanalysis models for the evolving wind energy sector.

**Response:**

We thank the reviewer and appreciate the positive feedback and constructive comments on our article. We address the comments below and updated the manuscript adequately.

**1.2 Specific comment**

> **Comment 1**
>
> Why is the FINO1 platform used for model validation when it is located near several wind farms? As noted in the manuscript, this proximity likely affects the measurements, making FINO1 unsuitable for validation unless the models explicitly account for the wind farms or the data are filtered to exclude disturbed wind directions. Since the measurements at FINO3 do not have nearby wind farms, wouldn't they already provide a more representative view of undisturbed offshore conditions?

**Response:**

For a complementary answer, we refer to our response to Reviewer 1. We consider it important to include the FINO1 data to illustrate how the presence of a wind farm may affect the agreement between modelled wind data and lidar measurements. Rather than dismissing the data, we believe it is valuable to highlight and discuss this interaction, as it documents how wind speed can be affected by man-made offshore structures. We believe that including both the FINO1 and FINO3 datasets enhances the value of our study, particularly as few previous studies, to our knowledge, incorporate data from both platforms. This dual perspective adds depth to our analysis and offers useful contributions to future research, particularly in the context of wind resource assessments in areas with sea-covered wind farms.

we have added the following lines in the manuscript

*The measurement data were collected by reference DWL instruments within the area covered by ERA5, NORA3, and NEWA. Two lidar campaigns were conducted in the marine ABL at the FINO1 and FINO3 locations, two others at coastal sites (Sola and Lista airports in Norway), and one in complex terrain (Bjerkreim, Norway). This diverse set of locations—comprising offshore, coastal, and complex terrain sites—provides a robust basis for assessing the performance of wind atlases for tall wind profiling. The FINO1 and FINO3 platforms offer complementary datasets for analyzing wind conditions near offshore energy installations (Podein et al., 2022). FINO1's proximity to wind farms enables an examination of discrepancies between lidar measurements and wind atlases. Combining data from both platforms highlights the challenges of wind resource assessment in such*

*areas and underscores the need for cautious application of wind atlases near off-shore energy projects.*

**Comment 2**

In line 300, it is mentioned that the EMD values are comparable across all models at coastal locations. However, this is not the case for the Sola site, where there are noticeable differences between the models.

**Response:**

We agree with the reviewer. This sentence seems to be an artefact of an older version of the manuscript. This sentence has been removed for the sake of brevity and the next sentence has been adjusted as

*At the coastal sites and complex terrain Bjerkeim, NORA3 achieves the lowest EMD, underlining its potential in heterogeneous topographies. As expected, ERA5 shows significantly higher EMD values than the other two models onshore, which is attributable to its lower horizontal spatial resolution.*

**Comment 3**

The paper emphasizes the validation of hindcast data at higher altitudes, beyond what has been extensively studied. Given this, why focus on results at 150 m, a height already typical for current wind turbines, when higher-altitude data are available? The higher- altitude comparisons would seem more aligned with the study's stated objectives.

**Response:**

We have added a new figure showing the bar plot at the altitude closest to 300 m and added the following lines in the manuscript to clarify the comment of the reviewer

*Figure 8 compares four error metrics describing the discrepancies between measurements and modelled mean wind speed data across five sites: FINO1, FINO3, Sola, Bjerkreim and Lista at a single height, corresponding to the range gate of the lidar nearest to 150 m. Figure 9 shows similar error metrics at a height of ca. 300 m, which is more relevant for airborne wind energy ssytems. The results presented in Figs. 8-9 highlight key differences in error metrics among the wind datasets, complementing the error metric profiles shown in Fig. 10. Notably, the variability in error metrics observed at 150 m and 300 m aligns with the one seen at other altitudes, reinforcing the consistency of our findings.*

The results at 150 m are representative of wind characteristics relevant to modern wind turbines, making them practical for wind energy applications and directly linked to the capacity factor analysis presented later. Additionally, the error metrics at 150 m complement the full-profile analysis and highlight key differences among the wind datasets at a reference height commonly used for model validation. While higher-altitude data are included in the profile analysis, focusing on 150 m also facilitates the discussion of turbine capacity factors later in the paper.

**1.3  Technical correction**

**Comment 4**

In the introduction, it might be appropriate to add the reference, where they use ERA5 to compute AEP of airborne wind energy systems: Schelbergen, M., Kalverla, P. C., Schmehl, R., and Watson, S. J.: Clustering wind profile shapes to estimate airborne wind energy production, Wind Energy Science, 5, 1097–1120, https://doi.org/10.5194/wes-5-1097- 2020, 2020

**Response:**

We agree with the reviewer; we have added this reference to the introduction when refering to the use of tall wind speed profile for wind resource assessment.

> **Comment 5**
>
> In line 54, the acronym "AWE" is repeated unnecessarily. Please use the acronym directly after the first mention.

**Response:**

We have adjusted the use of the acronym as suggested by the reviewer.

> **Comment 6**
>
> In line 111, it is generally not proper styling to add directly an url to the text. Please include it in the references and refer to that.

**Response:**

We follow the recommendation of the reviewer and we have now added the url link in the data availability section.

**References**

Podein, P., Tinz, B., Blender, R., & Detels, T. (2022). Reconstruction of annual mean wind speed statistics at 100 m height of FINO1 and FINO2 masts with reanalyses and the geostrophic wind. *Meteorologische Zeitschrift*, *31*, 89–100.

---

## Author Comment (AC2)

Responses to Reviewers' Comments for Manuscript
WES-2024-119

**Tall Wind Profile Validation Using Lidar Observations and Hindcast Data**

Addressed Comments for Publication to

**Wind Energy Science (ISSN 2045-2322)**

by

Cheynet et al.

**1 Authors' Response to Reviewer 1**

**1.1 General comment**

> **General Comments.** In this manuscript three model-based datasets (NORA3, NEWA and ERA5) are validated using Doppler wind lidar data obtained from five locations, including North Sea (FINO1, FINO3) and coastal and complex terrain locations in Norway. Emphasis is given to long-range scanning Doppler wind lidars, providing wind profiles far above the atmospheric surface layer that are relevant for modern wind turbine designs and airborne wind energy (AWE) systems. These altitudes that are not feasible for in- situ wind measurements in tall masts (especially offshore) or the more extensively used short-range Doppler lidar wind profilers that are limited to 200-300m. The validation focuses on altitudes between 100m and 500m, using various error metrics, and their corresponding capacity factors, based on power curves for various wind turbines and AWE systems. The authors claim an increasing agreement between the models and the measurements with height, and argue that those models are valuable R&D on AWE systems.
>
> In general, the manuscript addresses an important point, namely the need to validate models at altitudes relevant for future wind energy systems, and the lack of continuous, long-term measurement campaigns to do so. The authors point to the need of dedicated Doppler wind lidar profilers with sufficient height range, which are indeed lacking commercially right now. The manuscript also highlights the complexity of comparing the performance of various models, and that the best choice really depends on their actual application (type of location, relevant height range, . . . ).

**Response:**

We thank the reviewer and appreciate the positive feedback and constructive comments on our article. We address the comments below and updated the manuscript adequately.

**Comment 1**

I have some objection to the term "tall wind profile". Tall is used for physical structures, like masts or wind turbines, but a wind profile cannot be tall. I am not aware that "tall wind profile" is a commonly used term in our community, however, if I am wrong in this (i.e. it is used in more papers), I will drop my objection.

**Response:**

We acknowledge the reviewer's concern regarding the term "tall wind profile" and agree that it is important to use precise terminology. There is a need for a term to describe wind speed profiling over several hundred metres, distinguishing it from traditional wind profiling. While no universally accepted label exists for this specific context, the term "tall wind profile" has been used in boundary-layer meteorology as by Peña et al. (2014) and Kelly et al. (2014). Given this precedent, we believe the term is appropriate for the present case.

We have updated the following sentence to the manuscript: *Tall wind profiles, as defined here, cover the entire atmospheric boundary layer (ABL) or at least the initial 500 m above the surface. The term 'tall wind profile' is in line with its use in boundary-layer meteorology (e.g. Peña et al., 2014; Kelly et al., 2014).*

**Comment 2**

The FINO1 measurements are not suitable for validation due to the presence of an operating wind farm (and the models do not include that). Therefore it should be not be included here, also because FIN03 is already available to cover the offshore situation. Only if the data could be filtered to minimize the effect of the wind farm (for instance, if the influence is only present for certain wind directions), its inclusion would make sense.

**Response:**

The reviewer raises a valid point about the potential influence of the surrounding wind farm on the FINO1 data. However, we believe it is important to include the FINO1 data to demonstrate how the presence of a wind farm can impact its

agreement with the lidar data. This issue is explicitly addressed in the manuscript, as it highlights a key takeaway: wind atlases should be applied with caution in areas near existing wind farms. We believe that incorporating both the FINO1 and FINO3 datasets adds significant value, particularly as, to our knowledge, few studies have utilized both. While the suggestion to filter the data to minimize the wind farm's influence could serve as a topic for a master's thesis, such an analysis lies beyond the scope of this study.

we have added the following lines in the manuscript

*The measurement data were collected by reference DWL instruments within the area covered by ERA5, NORA3, and NEWA. Two lidar campaigns were conducted in the marine ABL at the FINO1 and FINO3 locations, two others at coastal sites (Sola and Lista airports in Norway), and one in complex terrain (Bjerkreim, Norway). This diverse set of locations—comprising offshore, coastal, and complex terrain sites—provides a robust basis for assessing the performance of wind atlases for tall wind profiling. The FINO1 and FINO3 platforms offer complementary datasets for analyzing wind conditions near offshore energy installations (Podein et al., 2022). FINO1's proximity to wind farms enables an examination of discrepancies between lidar measurements and wind atlases. Combining data from both platforms highlights the challenges of wind resource assessment in such areas and underscores the need for cautious application of wind atlases near offshore energy projects.*

> ### Comment 3
>
> The authors note that the conclusions on the model performances for the different sites might be hampered by the quality of the different Doppler lidar instruments. However, those measurements have been validated with other measurements, as described in Section 2.2. Wouldn't it therefore not be possible to quantity whether the validation results are significant in terms of the measurement uncertainty or bias?

**Response:**

The performance of the lidar can vary over time due to calibration drift, maintenance needs, and transport-induced shocks. Although systematic calibration

could, in theory, ensure consistent performance across WindCube 100S units, practical constraints like deployment schedules make this difficult. These factors introduce uncertainties that cannot be fully quantified in this study.

**Comment 4**

For the wind measurements at the relevant altitudes the authors immediately jump to Doppler lidar in the introduction. But there are also other remote sensing instruments that can measure wind between 200 and 500m, like radar wind profilers and sodar. In fact, for this validation study, their temporal and vertical resolution would be more than sufficient. The choice of Doppler lidar should be given a bit more context and motivation.

**Response:**

We agree with the reviewer. We have added the following lines in the introduction:

*Tall wind speed profiles can in general be measured using remote sensing technologies (Emeis, 2011), including Doppler wind radar (Lehmann & Brown, 2021), sodar (Bianco, 2011), and lidar (Pichugina et al., 2012). As commercial Doppler wind lidars (DWLs) have become the standard instrumentation for wind energy applications, we have based our study on corresponding available lidar data sets.*

**Comment 5**

The conclusion that there is an increasing agreement between models and lidar measurements, as stated in the abstract, is not explicitly stated in the main text, including the conclusion. Either the main text is underselling the results, or the abstract is overselling it.

**Response:**

Thank you for pointing this out. An increasing agreement between models and lidar measurements is indeed a finding of this study. We have updated the manuscript

to explicitly emphasize the increasing agreement between models and lidar measurements in the main text, while being fairly nuanced. We have reformulated this sentence in the abstarct. The following sentence has been added in the main text:

*In most cases, the agreement between the models and lidar measurements improves with height. However, this trend is not consistently observed at coastal and complex terrain sites, where deviations can occur, depending on the error metric and model database.*

and in the conclusion:

*While the agreement between the models and lidar measurements generally improves with height, this trend is less consistent at coastal and complex terrain sites, where deviations occur, especially for ERA5 and NEWA.*

**Comment 6**

I was a bit surprised that although the paper emphasis the need for wind profile beyond what can be reached by traditional masts and (floating) short-range wind lidars, still most of the presented results are at an altitude of 150m (for which, by the way, there are much more lidar data available, including offshore). Why this particular choice?

**Response:**

The choice of 150 m reflects the hub height of the 15 MW wind turbine analysed in Fig. 10, enabling a more direct connection between the wind profile results and the turbine's capacity factor. Additionally, Fig. 7 should be interpreted alongside Fig. 8, which provides the profiles of the error metrics to offer a more comprehensive view of the model's performance across altitudes. We acknowledge that the focus on this altitude might seem limiting given the availability of lidar data at similar heights, but we believe that it is relevant for the context of this study (offshore wind energy). In the revised manuscript, we include a similar bar plot at a height of 300 m, which may be more relevant for airborne wind energy systems. However, merging these two figures was not feasible, as we prioritized achieving a balance between clarity and the concise presentation of information.

**1.2 Specific comments**

**Comment 7**

Title: "validation using lidar observations and hindcast data". Are you not validating hindcast data using lidar observations?

**Response:**

This is correct. Following the comments from the other reviewers, we have reformulated the title as "Tall wind profile validation of ERA5, NORA3 and NEWA, using Lidar observations"

**Comment 8**

Table 1, why this table is in the manuscript? To make the point that there are very limited amount of tall towers with in-situ wind measurements, such a table is not required.

**Response:**

The table serves two purposes: (1) to illustrate the scarcity of tall masts with wind measurements, and more importantly, (2) to document where these masts are located and provide a resource for readers seeking more information. We believe this table offers valuable context for the study and helps orient readers, especially those interested in understanding the availability and characteristics of these rare measurement sites. None of the other reviewers raised objections to the inclusion of the table.

**Comment 9**

Section 3.3: It is not explained how the wind profile is used in the calculation of CF for wind turbines. Is this wind speed at hub height taken or a rotor average. Table 4 provides hub height and rotor diameter of the various wind turbine types, but nowhere it written how this information is used. This is in contrast to the extension discussion on the AWE system.

**Response:**

We agree that this aspect should be clarified. We used the wind speed at hub height rather than rotor-averaged wind speed for the calculation of capacity factors (CF). While rotor-averaged (or rotor-equivalent) wind speed, which accounts for effects such as shear, turbulence intensity, and wind veer (Wagner et al., 2009; Antoniou et al., 2009; Murphy et al., 2019), could provide more accurate capacity factor estimates, this level of detail is beyond the scope of the current study. Assessing its impact would ideally require output data from a full-scale large offshore wind turbine for validation. To clarify this, we have added the following lines to the manuscript:

*In this study, capacity factor calculations are based on wind speed at hub height. Modelling the rotor-averaged (or equivalent) wind speed, which accounts for shear, turbulence intensity, and wind veering (Wagner et al., 2009; Antoniou et al., 2009; Murphy et al., 2019) may yield more realistic capacity factor estimates and further justify the modelling of tall wind profiles for the design of large offshore wind turbines. However, such an analysis falls beyond the scope of the present work.*

**Comment 10**

In correct usage of term "In-situ" throughout the manuscript. Doppler lidar is a remote- sensing instrument and definitely not "in-situ"! However, in distinguishing between model and measurement data, in several parts of the manuscript the term "in-situ" is used for Doppler lidar, which is wrong. This needs to be corrected.

**Response:**

We agree with the reviewer and have removed the term "in-situ" throughout the manuscript, as it is not necessary and partly redundant.
* * *
**Comment 11**

The distinction coastal and complex locations from Figure 2 is not clear (at least for the non- Norwegian reader). Would a zoom-in of the map help to clarify the difference between the Sola and Lista as coastal/non-complex, and Bjerkeim as complex terrain?

**Response:**

We agree with the reviewer, we have added a topographic map of the area, which better highlight the difference between the sites as well as the following updated lines:

*Figure 2 summarises the locations and measurement periods of the five campaigns selected for the validation of wind atlases while figure 3 provides a close-up of the three onshore locations. The offshore sites are situated in open waters, whereas the coastal sites are only a few kilometres from the shore, characterised by sharp roughness changes as the terrain transitions from open water to flat, agricultural land with sparse vegetation. These abrupt roughness changes introduce an internal boundary layer, which can be challenging to capture accurately in hindcast and reanalysis wind speed models. The complex site is mountainous, with steep slopes and limited vegetation or trees. While distinct from the fjord-like landscapes found in other parts of Norway, the complex terrain features significant elevation changes that contribute to non-homogeneous wind conditions, particularly within the atmospheric surface layer.*
* * *
**Comment 12**

Section 5.2: Could you be more explicit, or give examples, on what you mean with "microscale models"

**Response:**

We have revised Section 5.2 and added the following explanation at the beginning of the section:

*In wind energy, wind simulations are typically performed using two types of models: mesoscale models, which provide wind speed data over spatial scales ranging from a few kilometres to hundreds of kilometres, and microscale models, which operate at smaller scales, from a few metres to approximately one kilometre. While these models are complementary, microscale models are particularly useful in capturing wind flow in complex terrain, where topographic features significantly influence wind conditions, or near structures such as buildings and wind farms.*

*This study primarily focuses on mesoscale-derived wind speed data, which can be limited in capturing fine-scale flow features in complex terrains or near coastal sites. For offshore sites like FINO3, microscale effects are likely negligible. However, for coastal sites such as Sola and Lista, and for the complex terrain at Bjerkreim, microscale modelling may enhance the agreement between simulated and measured wind speeds. At Bjerkeim, computational fluid dynamics (CFD) models could help capture complex phenomena, such as flow recirculation and detached downslope flow, which are prevalent in mountainous terrain like Southeastern Norway.*

*At FINO1, microscale flow simulations may also be needed to model wake effects on in-situ measurements. Future studies should investigate the benefits of coupling mesoscale and microscale models to enhance performance metrics at coastal and complex sites. We anticipate this coupling could shift the near-zero bias of NORA3 to slightly negative or positive values, while potentially reducing the current bias of NEWA and ERA5 towards zero. However, such an analysis lies outside the scope of the present study.*
* * *
**Comment 13**

Section 5.2: At the end of this section the issue of Doppler lidar wind profiling measurements in complex terrain is mentioned. This is a relevant point, but doesn't belong to this section (which is about the models). Maybe this issue should be discussed much earlier in the paper. Are there solutions to this issue, or would validation in complex terrain remain problematic?

**Response:**

We agree with the reviewer and have moved the discussion on Doppler lidar wind profiling measurements in complex terrain to just before Section 4.3, where it fits more appropriately. We have slightly shortened the content to better align with the context and added a reference to Klaas-Witt & Emeis (2022):

*The discrepancy between modelled wind speed data and lidar-based measurements at the complex site Bjerkreim and the coastal sites Sola and Lista is influenced by the higher occurrence of non-homogeneous flow fields at onshore sites compared to offshore locations. These effects, particularly within the first 300 m above the surface, can exacerbate the measurement uncertainties of lidar retrievals using DBS or velocity-azimuth display scanning (Klaas-Witt & Emeis, 2022).*

Higher measurement errors in complex terrain are a recognised challenge for Doppler wind lidar (DWL) profilers, as they rely on fundamental assumptions about homogeneous flow. However, advancements such as the use of a 5-beam DBS scanning mode, instead of the traditional 4-beam mode, have significantly improved DWL profiler performance in complex terrain for the past 10 years.

**References**

Antoniou, I., Pedersen, S. M., & Enevoldsen, P. B. (2009). Wind shear and uncertainties in power curve measurement and wind resources. *Wind Engineering*, *33*, 449–468.

Bianco, L. (2011). Introduction to sodar and rass-wind profiler radar systems. In D. Cimini, G. Visconti, & F. S. Marzano (Eds.), *Integrated Ground-Based Observing Systems* (pp. 89–105). Berlin, Heidelberg: Springer Berlin Heidelberg.

Emeis, S. (2011). *Surface-Based Remote Sensing of the Atmospheric Boundary Layer* volume 40 of *Atmospheric and Oceanographic Sciences Library*. Dordrecht: Springer Netherlands. URL: http://link.springer.com/10.1007/978-90-481-9340-0. doi:10.1007/978-90-481-9340-0.

Kelly, M., Troen, I., & Jørgensen, H. E. (2014). Weibull-k revisited:"tall" profiles and height variation of wind statistics. *Boundary-Layer Meteorology*, *152*, 107–124.

Klaas-Witt, T., & Emeis, S. (2022). The five main influencing factors for lidar errors in complex terrain. *Wind Energy Science*, *7*, 413–431.

Lehmann, V., & Brown, W. (2021). Radar wind profiler. In T. Foken (Ed.), *Springer Handbook of Atmospheric Measurements* (pp. 901–933). Cham: Springer International Publishing. URL: https://doi.org/10.1007/978-3-030-52171-4_31. doi:10.1007/978-3-030-52171-4_31.

Murphy, P., Lundquist, J. K., & Fleming, P. (2019). How wind speed shear and directional veer affect the power production of a megawatt-scale operational wind turbine. *Wind Energy Science Discussions*, *2019*, 1–46.

Peña, A., Floors, R., & Gryning, S.-E. (2014). The Høvsøre tall wind-profile experiment: a description of wind profile observations in the atmospheric boundary layer. *Boundary-layer meteorology*, *150*, 69–89.

Pichugina, Y. L., Banta, R. M., Brewer, W. A., Sandberg, S. P., & Hardesty, R. M. (2012). Doppler lidar–based wind-profile measurement system for offshore wind-energy and other marine boundary layer applications. *Journal of Applied Meteorology and Climatology*, *51*, 327–349.

Podein, P., Tinz, B., Blender, R., & Detels, T. (2022). Reconstruction of annual mean wind speed statistics at 100 m height of FINO1 and FINO2 masts with reanalyses and the geostrophic wind. *Meteorologische Zeitschrift*, *31*, 89–100.

Wagner, R., Antoniou, I., Pedersen, S. M., Courtney, M. S., & Jørgensen, H. E. (2009). The influence of the wind speed profile on wind turbine performance measurements. *Wind Energy: An International Journal for Progress and Applications in Wind Power Conversion Technology*, *12*, 348–362.

---

## Author Comment (AC3)

Responses to Reviewers' Comments for Manuscript
WES-2024-119

**Tall Wind Profile Validation Using Lidar Observations and Hindcast Data**

Addressed Comments for Publication to

Wind Energy Science (ISSN 2045-2322)

by

Cheynet et al.

**1  Authors' Response to Reviewer 3**

**1.1  General comment**

> **General Comments.** The submitted paper, "Tall Wind Profile Validation Using Lidar Observations and Hindcast Data" by Etienne Cheynet et al., analyses the accuracy of three different wind atlases (NEWA, NORA3 and ERA5) by comparing them to long-range wind lidar measurements. Various error metrics such as bias, RMSE, EMD, $R^2$ are evaluated for five different sites including offshore (FINO1, FINO3), coastal onshore (Sola, Lista) and complex terrain (Bjerkeim). The study focuses on wind conditions at heights between 100 m and 500 m, a range not feasible for traditional met masts or short-range lidar profilers, making it directly applicable to tall wind turbines and airborne wind energy (AWE) systems. Additionally, the authors assess the estimated capacity factors of reference wind turbines (NREL 5 MW, NREL 18 MW, and IEA 15 MW) and AWE systems (3 MW fixed-wing and 100 kW EnerKíte semi-rigid), comparing simulated and measured wind data as a further quality metric. The authors conclude that all three wind atlases perform well offshore, with NORA3 and ERA5 showing slightly better performance above 200 m. Onshore, NORA3 consistently outperforms ERA5 and NEWA at all heights, emphasizing that the choice of a suitable wind model depends on specific application, location, and height requirements. The paper addresses important topics, such as the adequate choice of wind data for initial wind resource assessment, validation of wind models at heights relevant for future wind energy systems, and the need to develop DWL profilers to reliably measure these heights over the long term.
>
> Here are some general comments I would like to see addressed before publishing the paper. However, some of these comments are a matter of personal preference, and I would appreciate hearing the authors' opinion if they choose not to implement them.

**Response:**

We sincerely thank the reviewer for their thorough and detailed review of the manuscript. We greatly appreciate the considerable effort and valuable insights, which we believe have significantly strengthened the quality of the manuscript.

**Comment 1**

Please rethink the title "Tall Wind Profile Validation Using Lidar Observations and Hindcast Data". This title sounds like you are validating tall wind profiles using lidar and hindcast data. I am not familiar with the term Hindcast, but it is my understanding that only NORA3 is Hindcast data, ERA5 is climate reanalysis data and NEWA derived from WRF and WAsP with boundary conditions from ERA5. Are you using Hindcast and reanalysis interchangeably? It would good if the title would reflect that you are validating and comparing lidar measurements with different wind models up to higher altitudes.

**Response:**

We agree with the reviewer. As stated in our reply to reviewer 1, we have now changed the title into: "Tall wind profile validation of ERA5, NORA3 and NEWA, using Lidar observations". Only the mesoscale ouput of NEWA were used as the microscale output are not available as time series.

**Comment 2**

Please clarify the writing. Several sentences are difficult to understand or can be understood in various ways. See attached commented document..

**Response:**

We have carefully revised the manuscript based on the detailed comments provided in the attached document to improve readability and reduce ambiguities.

**Comment 3**

Appendix: Why was the section moved to the appendix? It is my understanding that while this approach delivers good interpolation, it is not used because the improvement doesn't justify the increased afford? I think you could remove it for clarity and only focus on the approach you took, but I am open to hearing your opinion.

**Response:**

Yes, that is correct. We did not observe a substantial improvement with the more complex non-linear regression compared to the simpler and more robust linear interpolation. However, we chose to include these results in the appendix to document that alternative approaches were considered. We believe this adds transparency to our methodology and demonstrates that the chosen approach is not the only viable option. Additionally, presenting these results as figures in the appendix is more effective and concise than a textual explanation, as visualizations often provide clearer insights than lengthy descriptions.

**Comment 4**

You determine bias, $R^2$, RMSE and EMD only in terms of horizontal wind speed. Did you also investigate the directional difference between the models and measurements and are they meaningful or significant?

**Response:**

Yes, we did investigate directional differences between the models and measurements. In an earlier version of the manuscript, we included this information, replacing the EMD metric with the Circular EMD to discuss error metrics for the wind direction. However, the manuscript became excessively lengthy and heavy. Thus we had to remove this part to streamline the presentation. We acknowledge that wind direction, particularly veering, is an important topic for large offshore wind turbines and AWE systems. Nonetheless, we made the decision to focus on horizontal wind speed to keep the paper concise and accessible.

**Comment 5**

Wind data: Please clarify how you compared the model and measurement data. You mention a spatial and temporal interpolation of model data to the lidar location, height and time. How did you interpolate between 30 or 60 min modeled wind data to 10 min increments as the measurements? Is it a linear interpolation? Considering how quickly the wind changes that leads to significant differences. I think it would be better if you averaged the 10 min measurements to 30 min or 60 min instead.

**Response:**

Thank you for your comment. As explained in the manuscript, we tested both approaches: interpolating the model data to 10-minute intervals and downsampling lidar data to match the model's 30- or 60-minute resolution. The differences between the two methods were minimal. However, downsampling the lidar data introduced challenges, particularly due to irregular sampling (e.g., at FINO1, only two 10-minute scans were conducted per hour) and data gaps that required additional processing, such as gap-filling. These steps added complexity and potential errors. Given the limitations and the fact that 10-minute averaging is a standard in wind energy studies, we chose to interpolate the model data to 10-minute intervals for consistency and simplicity. While we acknowledge that this approach may slightly increase errors due to rapid wind changes, the alternative would also introduce errors from data processing and manipulation.

**Comment 6**

Measurement campaigns: The duration of measurement campaigns, particularly at Sola and Bjerkreim are very short and seems to have a lot of data missing (Figure: 6). Please comment on what the reason for this is and add a brief statement in Section 2.2. that these measurements are not representative of the typical, annual wind variations at these sites.

**Response:**

The measurements at Bjerkreim and Sola were conducted using early prototype versions of commercial Doppler wind lidars, which were less reliable than the

instruments available today. This led to significant data gaps and shorter campaign durations. Additionally, logistical constraints and the limited objectives of these campaigns contributed to their brevity. We have now added a statement in Section 2.2 to clarify that these measurements are not representative of the typical, annual wind variations at these sites. The added text reads:

*It should be noted that the measurements at Bjerkreim and Sola were conducted over short time periods and are therefore not representative of the typical, annual wind variations at these sites. Instead, the data should be interpreted within the context of this study, which aims to compare tall wind speed profiles from wind atlases with lidar observations.*
* * *
**Comment 7**

Why are you comparing wind data "at the range gate nearest to 150 m"? Why this height and not 200 m or 300 m which is closer to operating heights of tall wind turbines and AWEs? Why did you not interpolate to a specific height to compare them better?
* * *
**Response:**

To complement the error bar at 150 m, we have added a new figure showing similar error bars at 300 m. The operating height of tall wind turbine is typially the hub height (150 m) , although he tip top height can/will reach indeed 200-300 m. As elaborated in our reply to reviewer 2, we use the height of 150 m for consistency with the description fo the capacity factor. Note that Figure 8 shows the profiles of these error metrics, so the error metrics at the other heigths are still provided in the manuscript. we have added the following description for these figures:

*Figure 7 and Fig. 8 compare four error metrics describing the discrepancies between measurements and modelled mean wind speed data across the five sites at range gates closest to 150 m and 300 m, respectively. These results complement the profiles shown in Fig. 9. Notably, the variability in these metrics across the models observed in Fig. 7 and Fig. 8 aligns with trends at other altitudes, reinforcing the consistency of our findings.*

**Comment 8**

Please introduce the Taylor diagrams a bit more and the what conclusions you can draw from them.

**Response:**

We have expanded the description of Taylor diagrams to improve clarity in the method section, which reads as

*To complement these metrics, the Taylor diagram (Taylor, 2001) provides a summary of model performance by integrating the correlation coefficient, standard deviation, and RMSE into a single plot. This graphical representation is particularly useful for comparing multiple models against observed data in a visually intuitive way.*

**Comment 9**

Do you really need section 5? I think you could merge it with section 6, but I would like to hear your opinion too.

**Response:**

We recognize that there are different approaches to structuring academic papers, with some preferring to merge the discussion and conclusion sections, while others advocate for keeping them separate. Both approaches are acceptable, and in this case, we opted to keep them distinct for the sake of clarity.

**Comment 10**

Please spend a few sentences introducing the different AWE models. Introduce the system design, size, soft-kite, rigid-wing or semi-rigid wing, operating conditions, limitations and model assumptions.

**Response:**

We have reformulated the paragraph in the introduction presenting airborne wind energy (AWE) systems as below. Note that we must remain concise as the paper is not a review paper or focusing solely about AWE.

*Airborne Wind Energy (AWE) systems harness wind energy using tethered aircraft operating at altitudes between 200 and 600 m. At these heights, wind speeds are generally stronger and steadier than near the surface. Since the 2010s, AWE systems have made significant advances (Vermillion et al., 2021; Fagiano et al., 2022; Eijkelhof & Schmehl, 2022). Prototypes with capacities exceeding 600 kW have been developed, and scaling to multi-megawatt systems has been proposed (Vermillion et al., 2021; Kruijff & Ruiterkamp, 2018). Despite this progress, AWE systems are still in the early stages of development compared to offshore wind turbines. Two main concepts dominate current AWE designs. Ground-generation systems, or "pumping power" systems, generate energy on the ground using a winch and generator. The tethered aircraft alternates between energy-generation and recovery phases.*

*Aircraft for this concept include soft kites, semi-rigid wings, and rigid wings. Each type offers trade-offs between adaptability and durability. Onboard generation systems, in contrast, produce energy in the air using onboard turbines and power is transmitted to the ground via conductive tethers. These systems typically use rigid-wing aircraft, quadrotors, or toroidal aerostats (Cherubini et al., 2015). While ground-generation systems are relatively efficient, they require advanced automation for continuous operation (Elfert et al., 2024). Onboard-generation systems are better at harnessing high-altitude winds but face challenges in weight optimization and tether design. Flexible wings are adaptable to varying wind conditions but are less durable. Conversely, rigid wings provide higher power output but come with greater mechanical complexity and costs (Fagiano et al., 2022). Key challenges remain for AWE systems, including managing wind variability, tether dynamics, and autonomous operation. A major limitation lies in the reliance on oversimplified wind speed approximations, due to the lack of detailed wind speed data at altitudes above 200 m (Sommerfeld et al., 2019). Addressing this gap through tall wind profiling is essential for optimizing AWE system design and unlocking their full potential for large-scale deployment.*

**Comment 11**

Please revise the figures to improve readability and clarity (see comments in PDF document)!

**Response:**

We have addressed the reviewer's feedback to improve the readability and clarity of the figures. However, it is important to note that the figures were designed to align with the two-column format used by Wind Energy Science, which can result in a slightly unconventional layout in the single-column review format.

**Comment 12**

I believe that it is good practice not to have empty sections before a subsection title, e.g. Sections 3 or 5. Either remove the subsection titles or write a very brief summary of the section before the first subsection title.

**Response:**

We understand the viewpoint of the reviewer and have followed the recommendation where we found it relevant and useful to improve the manuscript. However, we have not applied this change in all cases, for the sake of conciseness and limit redundancy.

**Comment 13**

Try to formulate mote active voice sentences. Some are mentioned in the attached document.

**Response:**

We have followed the reviewer's recommendation and revised the manuscript addressing the specific instances highlighted in the attached document.

**Comment 14**

You can remove several unused abbreviations and introduce $z$ for height or $\bar{u}$ for average horizontal wind speed

**Response:**

Following the the reviewer's suggestion and have removed unused abbreviations where appropriate. We have used ABL for atmospheric boundary layer more consistently. OWT is now replaced by offshore wind turbines for clarity. The abbreviation "WRF" is also removed as it was used only once. We have also introduced $z$ for height and $\bar{u}$ for average horizontal wind speed as recommended:

*Hereinafter, z denotes the height in meters above the surface, and $\bar{u}$ represents the horizontally averaged mean wind speed at height z.*

**Comment 15**

Please add hyperlinks to the references, e.g. citation A, Tab.1, Fig.2, Eq. 3

**Response:**

We confirm that hyperlinks were used for reference and citation. We systematically use the cleveref and natbib packages in LaTeXwhich autmoatically format the reference style and create a hyperlink.

**Comment 16**

Capitalize "fig." and "table" in the entire the paper

**Response:**

We have adjusted the cleveref package to automatically capitalize references to figures and tables, following the reviewer's feedback.

**List of Main Changes in Response to the Reviewer's Comments**

We have carefully reviewed all comments and implemented most of the suggested changes. Below, we summarize key modifications and provide specific responses to certain points:

- **Acronyms:** We have re-evaluated the use of acronyms. Unnecessary acronyms have been removed, and we have not introduced any new ones to maintain clarity.

- **Figures:** Figures 2, 4, and 5–12 have been updated to address the reviewer's comments wherever possible. In a few cases, suggestions did not improve clarity, so we opted for a compromise. For example, please see Figure 10.

- **Table 4:** Table 4 has been removed as we now use only three turbine models to enhance clarity, making such a table unnecessary.

- **Terminology ("above the surface"):** We have reduced the use of the phrase "above the surface."

- **Use of transitional words:** We have improved the logical flow of the manuscript by revisiting transitional words (e.g., "therefore") throughout the text.

- **Definitions of "hindcast" and "reanalysis":** Brief definitions of hindcast and reanalysis are now provided.

- **Consistency in writing style:** We recognize that writing styles can differ. Both the reviewer's and our chosen style are valid, as long as consistency is maintained.

- **Abstract and acronyms:** We regard the abstract as a separate entity. Therefore, acronyms and definitions introduced in the abstract are reintroduced in the main text.

- **Introduction:** The last paragraph of the introduction has been clarified.

- **4D-Var data assimilation reference:** We have added a reference to 4D-Var data assimilation. We believe that the reference is more suitable than an extensive explanation, which would be beyond the scope of this study.

- **URLs and references:** URLs have been removed and replaced by corresponding BibTeX references.

- **Clarification of ASL:** The distinction between ASL (atmospheric surface layer) and "above sea surface" has been clarified. We have removed the acronym for "above sea surface."

- **HARMONIE-AROME definition:** HARMONIE-AROME is defined as a numerical weather prediction model in the revised manuscript. For comparison, the Weather Research and Forecasting (WRF) model is also a numerical weather prediction model.

- **ERA5 wind speed data:** ERA5 provides wind speed data at pressure levels and two height levels (10 m and 100 m). These levels do not always overlap. Combining both datasets adds robustness to our analysis.

- **Marine ABL terminology:** We prefer "marine ABL" over "offshore" because the former is more precise and indicates flow characteristics above the sea, whereas "offshore" may include coastal or seaside measurements.

- **Pressure-to-height conversion:** The conversion from pressure level to height level depends on atmospheric conditions and varies over time. This is why we use geopotential height, which also varies with time.

- **Angle conventions:** Angle conventions differ for scanning lidars (elevation relative to horizontal) and profilers (opening angle). Referring to "opening angle" for scanning lidars would be confusing, in our opinion.

- **Geopotential vs geometric height:** The difference between geopotential height and geometric height is negligible (close to or below 1 cm at 500 m above the surface). For brevity, we believe this aspect does not require further elaboration in the manuscript.

- **Placement of appendix A:** Appendix A remains appropriately placed, as it is not directly used in the methods or results sections. It is necessary to demonstrate that spatial linear interpolation on vertical levels is appropriate.

- **Panel labeling:** We have labelled the subpanels of Figs 7,9,8,11,12 and the figure in appendix with letters for clarity.

- **AWE system parameters:** For brevity, we have not elaborated on additional AWE system parameters (e.g., operating height, size), as they are not directly relevant to the results section.

- **Interpolation of wind data:** Whenever possible, we interpolate wind atlas data to the lidar range gate height. When not possible (e.g., for capacity factor calculations), we interpolate both wind atlas and lidar data to the operational or hub height. This approach avoids data overprocessing. We tried to clarify this point in the revised manuscript.

- **Turbine types for capacity factor:** We now use only three turbine types to improve the clarity of our capacity factor results.

- **Capacity factor insight:** We confirm that a capacity factor of 10–20% represents a significant drawback for intermittent wind energy systems, as such systems are unlikely to be economically viable.

- **Figures 11 and 12:** We have merged Figures 11 and 12 (those showing the CF of AWE systems) for conciseness.

- **Revisions to sections 4.3 and 5:** Sections 4.3 and 5 have been partially rewritten.

- **Numerical model resolution:** A higher spatial resolution does not always improve numerical model outputs. For example, ERA5 performs as well as NORA3 offshore and outperforms NEWA, which is unexpected but plausible. Overly high resolution can distort grid elements in terrain-following grids, introducing numerical errors.

- **Non-linear regression (appendix A):** The non-linear regression in Appendix A uses least-squares fitting. Including height levels at 750 m could alter the results slightly.

**References**

Cherubini, A., Papini, A., Vertechy, R., & Fontana, M. (2015). Airborne wind energy systems: A review of the technologies. *Renewable and Sustainable Energy Reviews*, *51*, 1461–1476.

Eijkelhof, D., & Schmehl, R. (2022). Six-degrees-of-freedom simulation model for future multi-megawatt airborne wind energy systems. *Renewable Energy*, *196*, 137–150.

Elfert, C., Göhlich, D., & Schmehl, R. (2024). Measurement of the turning behaviour of tethered membrane wings using automated flight manoeuvres. *Wind Energy Science*, *9*, 2261–2282. doi:10.5194/wes-9-2261-2024.

Fagiano, L., Quack, M., Bauer, F., Carnel, L., & Oland, E. (2022). Autonomous airborne wind energy systems: accomplishments and challenges. *Annual Review of Control, Robotics, and Autonomous Systems*, *5*, 603–631.

Kruijff, M., & Ruiterkamp, R. (2018). A Roadmap Towards Airborne Wind Energy in the Utility Sector. In *Airborne Wind Energy: Advances in Technology Development and Research* (pp. 643–662). Singapore: Springer Singapore. doi:10.1007/978-981-10-1947-0_26.

Sommerfeld, M., Crawford, C., Monahan, A., & Bastigkeit, I. (2019). LiDAR-based characterization of mid-altitude wind conditions for airborne wind energy systems. *Wind Energy*, *22*, 1101–1120.

Taylor, K. E. (2001). Summarizing multiple aspects of model performance in a single diagram. *Journal of geophysical research: atmospheres*, *106*, 7183–7192.

Vermillion, C., Cobb, M., Fagiano, L., Leuthold, R., Diehl, M., Smith, R. S., Wood, T. A., Rapp, S., Schmehl, R., Olinger, D. et al. (2021). Electricity in the air: Insights from two decades of advanced control research and experimental flight testing of airborne wind energy systems. *Annual Reviews in Control*, *52*, 330–357.

---

## Referee Report (RR1)

[referee-annotated manuscript omitted]

---

## Author Response (AR2)

Responses to Reviewers' Comments for Manuscript
WES-2024-119

**Tall Wind Profile Validation Using Lidar Observations and Hindcast Data**

Addressed Comments for Publication to

Wind Energy Science (ISSN 2045-2322)

by

Cheynet et al.

**1 Authors' Response to Reviewer 3**

We thank the reviewer for their additional valuable feedbacks on our manuscript. We have read carefully the annotated pdf and updated the manuscript. Changes are visible in the marked-up manuscript. We have also detailed our answer to some key feedbacks below.

**Comment 1**

Some sections of the paper are written in very careful / vague language: could improve, might perform better... I think you could write well known and generally accepted facts in a straight forward way without needing to add additional references.

**Response:**
We acknowledge that we frequently use the conditional form. This is intentional, as many statements involve hypotheses, and using the indicative form without definitive proof could be misleading. Additionally, widely accepted facts sometimes evolve as new evidence emerges. For instance, our statement that "CFD models could help capture complex phenomena in complex terrain" reflects this cautious approach. It is often stated that CFD methods, including RANS and LES, can model wind conditions in complex terrain. However, in-situ comparisons have revealed significant limitations. This is also one reason why high-resolution wind tunnel studies of flow in complex terrain are still used nowadays as complementary tool. By using the conditional form, we aim to encourage careful interpretation of our statements.

**Comment 2**

Many sentences would benefit from active voice rephrasing.

**Response:**
We have revised the manuscript to replace passive constructions with active voice wherever appropriate.

**Comment 3**

One thing to keep in mind with LiDAR measurements is that the devices should be verified against met mast measurements, which is of course impossible for devices that measure 500m+.

**Response:**

We agree with the reviewer. Fortunately, the use of ground-based remote sensing devices, including LiDAR, for wind resource assessment is increasingly being standardized, at least for profiler lidar. We have added the following lines in the conclusion:

*Finally, it should be noted that wind speed profiles established by DWLs are typically validated against anemometers mounted on met masts. However, such comparisons become impractical at altitudes above 200 m. Consequently, the accuracy of wind speed profiles at these heights, as measured by profiler lidar or scanning lidar in profiler mode, requires further evaluation.*

**Comment 4**

Please double check your references. For example, many reference do not include any DOI where there are DOI readily available online.

**Response:**

Thank you for pointing this out. We have reviewed all references and added DOIs where available. Additionally, we corrected capitalization inconsistencies.

**Comment 5**

Double check the spell for the use of British spelling and American spelling, e.g. metre vs meter, analyse VS analyze

**Response:**

Good point! We have proofread the manuscript to ensure consistency with British English. However, we have retained "levelized" with a *z*, as the spelling with *s* appears to be uncommon.

> **Comment 6**
>
> Page 6, Lines 145: "as reflected in the database status at the end of 2024.": what does this mean

**Response:**

The database is continuously updated and at the end of 2024, the database extended from 1961 onward. In the future, this may not be the case any longer and the database could start in the 1940s, as done for the ERA5 database.

---

## Author Response (AR3)

**Reply to the editor – technical corrections**

**Etienne Cheynet, on behalf of the authors**

Title: I think it would be helpful for readers if you add "datasets" after "NEWA" because not everyone will know these abbreviations. In the abstract, you explicitly denote ERA5, NORA3 and NEWA as datasets, so using this in the title would make sense.

Thank you for the suggestion. I have accepted it. The title reads now as *Tall wind profile validation of ERA5, NORA3 and NEWA datasets using lidar observations*

Line 16: "Airborne Wind Energy (AWE) systems" should be "Airborne wind energy (AWE) systems", as you also write in the abstract.

We have updated line 16, so that it reads as Airborne wind energy  (without captialisation of "wind" and "energy")

Lne 19: You write "Prototypes with capacities exceeding 600 kW have been developed" which is not correct. The largest flying prototype to date was Makani's M600, which had a capacity of just 600 kW (although this power was never achieved during operational testing, as far as I know). I would change "exceeding" into "up to", to be on the safe side.

Thank you for the correction. I have updated the sentence as "Prototypes with capacities up to 600 kW have been developed"

Line 28: The reference Elfert et al. (2024) is not the best choice to support the statement that ground-generation systems require advanced automation for continuous operation. There was no pumping cycle operation involved in this research, as it describes a tow test setup for the aerodynamic characterization of kites. Instead, I recommend the earlier references used by Vermillion et al. (2021) and/or Fagiano et al. (2022). Or, if you want more specialized ones, also Fechner & Schmehl (2018), https://doi.org/10.1007/978-981-10-1947-0_15, and Rapp et. al (2019), https://doi.org/10.2514/6.2019-1419, would do.

Thank you for the suggestions. I have replaced the reference to Elfert et al. (2024) with references to Vermillion et al. (2021) , Fagiano et al. (2022) and Fechner & Schmehl (2018). I had no access to the paper by Rapp et. Al (2019), so I did not include them but I assume three references are more than enough here :
*While ground-generation systems are relatively efficient, they require advanced automation for continuous operation \citep{Fechner2018,vermillion2021electricity,fagiano2022autonomous}.*

Lines 303 and 304: "eq. (2)" and "eq. (3)" should be "Eq. (2)" and "Eq. (3)" as per the style guide. Please check other uses.

I have updated the reference style using the cleveref package (\crefname{equation}{Eq.}{Eqs.}), which should update all equation references to follow the style guide.

Line 319: "Fig. 7" at the start of the sentence should be written out, i.e. "Figure 7" (see style guide). Also, when you reference two figures as you do here, "Fig. 7 and Fig. 8" (and elsewhere), it is better to combine this to the plural form "Figs. 7 and 8". You are doing this in some places, but in others, you don't.

This has now been adjusted. While checking the figures, we saw that Fig 4 was described before Fig 3. So I have now swapped their place. I have also added the Zenodo repository for the wind speed data: *The dataset used to generate the figures in this study is available on Zenodo (\url{https://doi.org/10.5281/zenodo.14848924}) under a BSD-3 open-access license.*